# Centrality-Dependent Chemical Potentials of Light Hadrons and Quarks Based on $p_T$ Spectrum and Particle Yield Ratio in Au-Au Collisions at RHIC Energies

Xing-Wei He [1], Hua-Rong Wei [1,*], Bi-Hai Hong [1], Hong-Yu Wu [1], Wei-Ting Zhu [1] and Feng-Min Wu [2]

1  Institute of Optoelectronic Technology, Lishui University, Lishui 323000, China
2  Department of Physics, Zhejiang Sci-Tech University, Hangzhou 310000, China
*  Correspondence: huarongwei@lsu.edu.cn

**Abstract:** We analyze the $p_T$ spectra of $\pi^{\pm}$, $K^{\pm}$, $p$, and $\bar{p}$ produced in different centralities' Au-Au collisions at different collision energies from 7.7 to 62.4 GeV using a two-component Erlang distribution in the framework of a multi-source thermal model. The fitting results are consistent with the experimental data, and the yield ratios of negative to positive particles are obtained from the normalization constants. Based on the yield ratios, the chemical potentials of light hadrons ($\pi$, $K$, and $p$) and quarks ($u$, $d$, and $s$) are extracted. This study shows that only the yield ratios of $p$ decrease with the increase in centrality. The logarithms of these yield ratios in the same centrality show obvious linear dependence on $1/\sqrt{s_{NN}}$. The extracted chemical potentials (the absolute magnitude for $\pi$) of light hadrons and quarks decrease with the increase in energy. The curves of chemical potential vs. energy for all centralities derived from the linear fits of the logarithms of the yield ratio as a function of energy have their maximum (the absolute magnitude for $\pi$) at the same energy of 3.526 GeV, which is possibly the critical energy of phase transition from a liquid-like hadron state to a gas-like quark state in the collision system.

**Keywords:** transverse momentum spectra; yield ratio of negative to positive particles; chemical potential of particle; critical end point of phase transition

## 1. Introduction

The successful running of the Relativistic Heavy Ion Collider (RHIC) in 2000 and the Large Hadron Collider (LHC) in 2008 [1] attracted great interest in studying the evolution process of interacting systems in high-energy collision. A large amount of evidence confirms that such a high-energy collision system produces an extremely high temperature and high density environment, which makes the collision system experience the phase transition process from the hadronic matter to quark–gluon plasma (QGP) [2–4], and produces strong coupling quark–gluon plasma (sQGP) [5–7]. By studying the chemical freeze-out temperature ($T_{ch}$) of the interacting system and the chemical potential ($\mu_B$) of baryon in the phase diagram of quantum chromodynamics (QCD) [8,9], one can obtain the information about the phase transition from hadronic matter to QGP and the properties of QGP, such as the possible critical end point (CEP) of phase transition [10,11]. Thus, it is important to study the baryon chemical potential in the $\mu_B - T_{ch}$ plane. Meanwhile, the chemical potentials of other particles, such as light hadron and quark, are also important and interesting in researching the evolution of collision systems, the mechanism of particle production and even the property of QGP. The final-state particles produced in high-energy collisions are multifarious and show many statistical behaviors that contain some information about the collision process [12–15]. It is interesting to find some useful information from these regular behaviors. After the kinetic freeze-out, the transverse momentum ($p_T$) of particle no longer changes [16]. Thus, by analyzing the $p_T$ distribution of final-state particles, one can obtain information about kinetic freeze-out, even other stages of collision system. For

example, one can extract the kinetic freeze-out temperature of the interacting system, the flow velocity of particles and so on, directly from the $p_T$ distribution [17]. One can also extract the yield ratio of negative to positive particles based on the $p_T$ spectrum and the chemical potential of particles at the chemical freeze-out [18–21]. Meanwhile, one can also analyze the connections between these quantities and the collision size, energy, centrality, particle mass, and so on, then further extract a particle production mechanism and the information of other earlier stages.

Generally, one can use phenomenological models to describe the $p_T$ spectrum of final-state particles. These models can be divided into microcosmic kinetics models and thermal statistical models. A thermal statistical model focuses on studying the collective or global statistical behavior of final-state particles. There are many theoretical distribution models in the framework of a thermal statistical model, such as Boltzmann distribution [22], blast-wave model [23], power-law function [24], Lévy distribution [25], Erlang distribution [17,26,27] and so on. In the framework of a multi-source thermal model [27–29], one can use the multi-component distribution model to improve the fitting degree of single-component distribution in the high $p_T$ region. Meanwhile, more information can be extracted. For example, by using multi-component Erlang distribution, one can not only extract the relative yield of particles, but also the weight of hard (soft) excitation degree.

From the yield ratio of negative to positive particles, one can obtain the chemical potentials of hadrons and quarks at chemical freeze-out, according to reference [30]. While the yield ratio calculated from $p_T$ spectrum of final-state particles, is actually at kinetic freeze-out when the yield ratio is affected by the strong decay from high-mass resonances and the weak decay from heavy flavor hadrons [31]. In order to obtain the yield ratio at chemical freeze-out, the contributions of strong decay and weak decay need to be removed from the yield ratio calculated from the $p_T$ spectra. While according to the references [21,32], the strong and weak decays actually have less effect on the above particle yield ratio from normalization constants, although they have a big impact on particle yield. As such, we can approximately extract the chemical potentials of hadrons and quarks by using the yield ratio from normalization constants instead of the yield ratio modified by removing the contributions of strong and weak decays.

In the present work, we describe the $p_T$ spectra of $\pi^\pm$, $K^\pm$, $p$, and $\bar{p}$ produced in different centralities' Au-Au collisions over a center-of-mass energy ($\sqrt{s_{NN}}$) ranging from 7.7 to 62.4 GeV [33,34] using a two-component Erlang distribution [17,26,27] in the framework of a multi-source thermal model [27–29]. The energy- and centrality-dependent yield ratios of negative to positive particles were obtained according to the extracted normalization constants. Meanwhile, the energy- and centrality-dependent chemical potentials of light hadrons ($\pi$, $K$, and $p$) and quarks ($u$, $d$, and $s$) are then extracted from the yield ratios.

## 2. The Model and Formulism

In the present work, we used a two-component Erlang distribution [26,27] to describe the $p_T$ spectra of the final-state light flavor particles to obtain the normalization constants, and to extract the yield ratios . The two-component Erlang distribution is regarded as the contribution of the soft excitation process and the hard scattering process. The soft excitation process comes from the interactions among a few sea quarks and gluons and results in the low-$p_T$ region distribution, and the hard scattering process originates from a harder head-on scattering between two valence quarks and results in the high-$p_T$ distribution. The two-component distribution is in the framework of a multi-source thermal model [27–29], and the method is as follows.

The multi-source thermal model assumes that many emission sources are formed in high energy-collisions. Due to the existence of different interacting mechanisms in the collisions and different event samples in experiment measurements, these emission sources are classified into $l$ groups. According to a thermodynamic system, the $p_T$ of particles generated from one emission source obey an exponential distribution,

$$f_{ij}(p_{tij}) = \frac{1}{\langle p_{tij} \rangle} \exp\left[ - \frac{p_{tij}}{\langle p_{tij} \rangle} \right], \tag{1}$$

where $p_{tij}$ and $\langle p_{tij} \rangle$ are the $p_T$ of particles from the $i$-th source in the $j$-th group and the mean value of $p_{tij}$, respectively. Assume that the mean $p_T$ of particles from each source in the same group is the same. Then, all the sources in the $j$-th group meet the distribution of the folding result of exponential distribution

$$f_j(p_T) = \frac{p_T^{m_j - 1}}{(m_j - 1)! \langle p_{tij} \rangle^{m_j}} \exp\left[ - \frac{p_T}{\langle p_{tij} \rangle} \right], \tag{2}$$

where $m_j$ is the source number in the $j$-th group and $p_T$ denotes the $p_T$ of particles from $m_j$ sources, i.e.,

$$p_T = \sum_{i=1}^{m_j} p_{tij}. \tag{3}$$

This is the normalized Erlang distribution, which can describe the $p_T$ distribution of the particles from the sources in the same group because they have the same excitation degree and stay at a common local equilibrium state. The contribution of all emission sources in all groups can be expressed as

$$f(p_T) = \sum_{j=1}^{l} k_j f_j(p_T), \tag{4}$$

where $k_j$ is the relative weight of the $j$-th group sources and meets the normalization $\sum_{j=1}^{l} k_j = 1$. This is the multi-component Erlang distribution, which can describe the final-state $p_T$ distribution. Then, the two-component Erlang $p_T$ distribution can be written as

$$f(p_T) = k_1 f_1(p_T) + (1 - k_1) f_2(p_T). \tag{5}$$

According to the two-component Erlang $p_T$ distribution, we describe the $p_T$ spectra of $\pi^{\pm}$, $K^{\pm}$, $p$ and $\bar{p}$ produced in Au-Au collisions at different energies for different centralities, and obtain the normalization constants corresponding to the above particles. The ratios of normalization constants of antiparticles, $\pi^-$, $K^-$, and $\bar{p}$, to particles, $\pi^+$, $K^+$, and $p$, are the yield ratios of negative to positive particles at kinetic freeze-out. Neglecting the little contribution of the strong and weak decays to the yield ratios, the ratios of normalization constants are approximately equal to the yield ratios of particles at chemical freeze-out. Due to the fact that the experimental data of some particles correspond to a narrow $p_T$ range, the normalization constant extracted by describing the $p_T$ spectra of particles with two-component Erlang distribution may be more precise than the yield published by the Collaborations.

Based on the above yield ratios, we calculated the chemical potentials of some light hadrons ($\pi$, $K$, and $p$) and light quarks ($u$, $d$, and $s$). According to the statistical arguments based on the chemical and thermal equilibrium within the thermal and statistical model [35], the three types of yield ratios, $k_\pi$, $k_K$, and $k_p$, in terms of the light hadron chemical potentials, $\mu_\pi$, $\mu_K$, and $\mu_p$, of hadrons $\pi$, $K$, and $p$, are to be [19,35,36]

$$k_\pi = \exp\left( - \frac{2\mu_\pi}{T_{ch}} \right),$$

$$k_K = \exp\left( - \frac{2\mu_K}{T_{ch}} \right),$$

$$k_p = \exp\left( - \frac{2\mu_p}{T_{ch}} \right), \tag{6}$$

where $T_{ch}$ is the chemical freeze-out temperature of the interacting system. Within the framework of a statistical thermal model of non-interacting gas particles with the assumption of standard Maxwell–Boltzmann statistics [2,3,37], $T_{ch}$ can be empirically obtained by the following formula

$$T_{ch} = T_{\lim} \frac{1}{1 + \exp[2.60 - \ln(\sqrt{s_{NN}})/0.45]},$$

(7)

where $T_{\lim}$ is the 'limiting' temperature and can be empirically taken to have a value of 0.164 GeV, and $\sqrt{s_{NN}}$ is in the unit of GeV [37,38].

Based on Equation (6) and references [19,39,40] under the same value of chemical freeze-out temperature for $\pi$, $K$, and $p$, we can obtain the three types of yield ratios in terms of the three types of quark chemical potentials ($\mu_u$, $\mu_d$, and $\mu_s$ for $u$, $d$, and $s$ quarks, respectively) to be

$$k_\pi = \exp\left[-\frac{(\mu_u - \mu_d)}{T_{ch}}\right] \Big/ \exp\left[\frac{(\mu_u - \mu_d)}{T_{ch}}\right] = \exp\left[-\frac{2(\mu_u - \mu_d)}{T_{ch}}\right],$$

$$k_K = \exp\left[-\frac{(\mu_u - \mu_s)}{T_{ch}}\right] \Big/ \exp\left[\frac{(\mu_u - \mu_s)}{T_{ch}}\right] = \exp\left[-\frac{2(\mu_u - \mu_s)}{T_{ch}}\right],$$

$$k_p = \exp\left[-\frac{(2\mu_u + \mu_d)}{T_{ch}}\right] \Big/ \exp\left[\frac{(2\mu_u + \mu_d)}{T_{ch}}\right] = \exp\left[-\frac{2(2\mu_u + \mu_d)}{T_{ch}}\right].$$

(8)

According to Equations (6) and (8), the chemical potentials of the above hadrons and quarks in terms of yield ratios can be, respectively, expressed as

$$\mu_\pi = -\frac{1}{2}T_{ch} \cdot \ln(k_\pi),$$

$$\mu_K = -\frac{1}{2}T_{ch} \cdot \ln(k_K),$$

$$\mu_p = -\frac{1}{2}T_{ch} \cdot \ln(k_p),$$

(9)

and

$$\mu_u = -\frac{1}{6}T_{ch} \cdot \ln(k_\pi \cdot k_p),$$

$$\mu_d = -\frac{1}{6}T_{ch} \cdot \ln(k_\pi^{-2} \cdot k_p),$$

$$\mu_s = -\frac{1}{6}T_{ch} \cdot \ln(k_\pi \cdot k_K^{-3} \cdot k_p).$$

(10)

In the present work, we only calculate the chemical potentials of the light hadrons of $\pi$, $K$, and $p$, and light quarks of $u$, $d$, and $s$. For the hadrons containing $c$ or $b$ quark, considering the fact that there is a lack of experimental data for the $p_T$ spectra which continuously vary with energy or centrality, we do not calculate the chemical potentials of $c$ and $b$ quarks, and the hadrons containing $c$ or $b$ quark. In addition, due to the lifetimes of the hadrons containing $t$ quark being too short to measure, we also cannot obtain the chemical potentials of $t$ quark, and the hadrons containing $t$ quark.

The above method is different from the conventional Hadron resonance gas (HRG) model [8]. Both baryo-chemical potential ($\mu_B$) and $T_{ch}$ are contained in the grand-canonical partition function of the hadron resonance gas. $\mu_B$ and $T_{ch}$ are obtained by fitting the experimental hadron yield, which is directly collected from the international collaborations, to HRG Model. However, information about $T_{ch}$ cannot be obtained for the nonexistence of the temperature parameter in the two-component Erlang distribution which is mainly used to attract the accurate yield. With the aid of the empirical formula (Equation (7)) about $T_{ch}$, the chemical potential of light hadrons and quarks can be calculated in this work. As $\mu_p$ is a proxy of $\mu_B$, a comparison between them will be discussed in the next section.

## 3. Results and Discussion

Figure 1 shows the $p_T$ distributions of (a)(d) $\pi^{\pm}$, (b)(e) $K^{\pm}$, (c) $p$, and (f) $\bar{p}$ produced in Au-Au collisions at $\sqrt{s_{NN}} = 7.7$ GeV in different centrality classes of 0–5%, 5–10%, 10–20%, 20–30%, 30–40%, 40–50%, 50–60%, 60–70%, and 70–80%. Similarly, the $p_T$ spectra for the energies of 11.5, 19.6, 27, 39, and 62.4 GeV are presented in Figures 2–6, respectively. $dN/dy$ on the axis denotes the rapidity density. The symbols represent the experimental data recorded by the STAR Collaboration in the mid-rapidity range of $|y| < 0.1$ [33,34]. The spectra for all centralities are scaled by suitable factors for clarity. The uncertainties are statistical and systematically added in quadrature. The curves are our results calculated using the two-component Erlang distribution. It should be mentioned that the data with different centralities at same collision energy show a similar trend. The best fit of data to the two-component Erlang distribution is obtained according to the combination of the minimum of $\chi^2$ and the shape of the curves. However, in our previous work [21], only $\chi^2$ was considered. The values of free parameters ($m_1$, $p_{ti1}$, $k_1$, $m_2$, and $p_{ti2}$), normalization constant ($N_0$), and $\chi^2$ per degree of freedom ($\chi^2$/dof) corresponding to the two-component Erlang distribution for different energies are, respectively, listed in Tables 1–6, where the normalization constant is for comparison between curve and data. One can see that the two-component Erlang distribution can well describe the experimental data of the considered particles in Au-Au collisions at all energies for all centrality classes. The tables show that the values of $m_1$ correspond to a low-$p_T$ region for all particles at all energies in all centrality classes are 2, 3, or 4, and all $m_2$ corresponding to high-$p_T$ region are 2, which shows that the soft process originates from the interaction among 2, 3, or 4 sea quarks and gluons, and the hard process originates from a hard head-on scattering between two valence quarks. The values of the relative weight factor $k_1$ of the soft excitation process are more than 60%, which reflects that the soft excitation is the main excitation process. In addition, the normalization constant $N_0$ increases with increase in energy and centrality, and decreases with increase in particle mass.

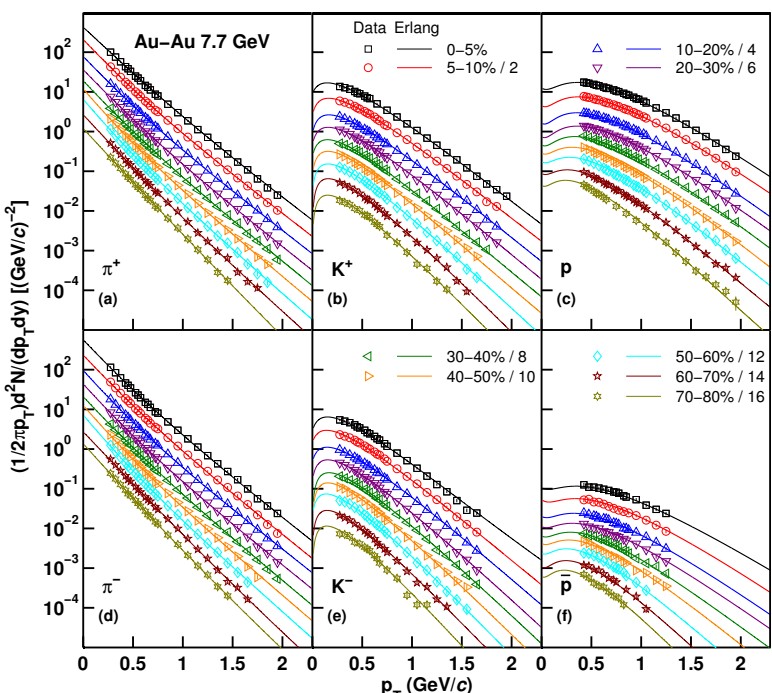

**Figure 1.** $p_T$ spectra for (**a**–**c**) positive ($\pi^+$, $K^+$, $p$) and (**d**–**f**) negative ($\pi^-$, $K^-$, $\bar{p}$) particles produced in Au-Au collisions with $|y| < 0.1$ at $\sqrt{s_{NN}} = 7.7$ GeV for different centralities (0–5%, 5–10%, 10–20%, 20–30%, 30–40%, 40–50%, 50–60%, 60–70%, and 70–80%). The experimental data represented by the symbols are measured by the STAR Collaboration [33]. The spectra for different centralities are scaled by suitable factors for clarity. The plotted error bars include both statistical and systematic uncertainties, and the curves are the two-component Erlang distribution fits to the spectra.

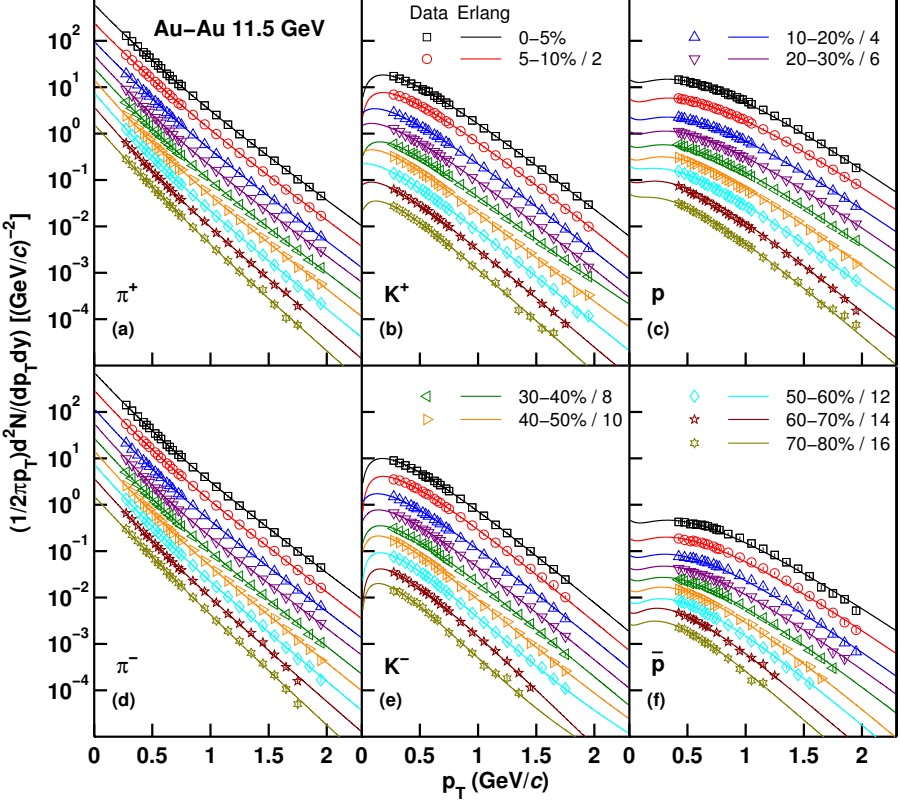

**Figure 2.** Same as Figure 1 but for Au-Au collisions at $\sqrt{s_{NN}} = 11.5$ GeV. The experimental data are from reference [33].

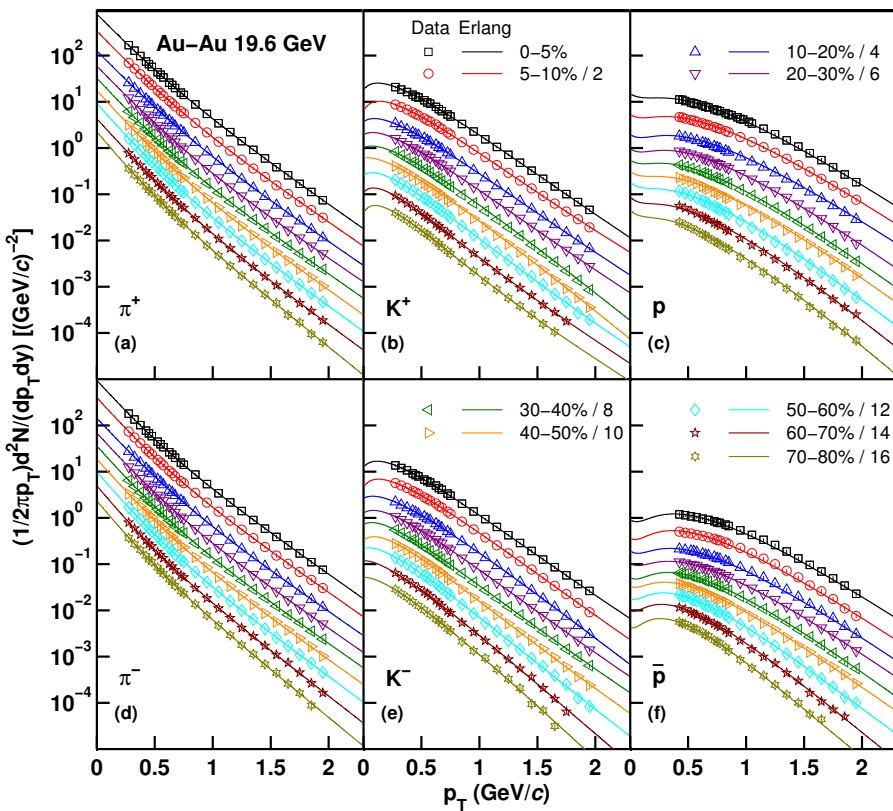

**Figure 3.** Same as in Figure 1 for Au-Au collisions at $\sqrt{s_{NN}} = 19.6$ GeV. The experimental data are from reference [33].

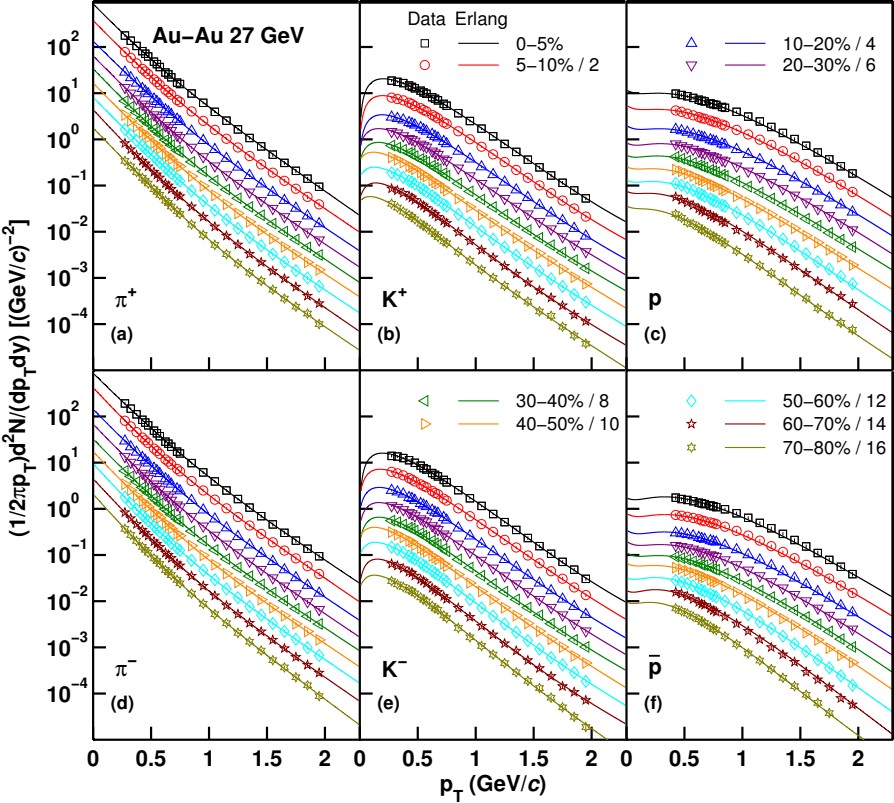

**Figure 4.** Same as Figure 1 for Au-Au collisions at $\sqrt{s_{NN}} = 27$ GeV. The experimental data are from reference [33].

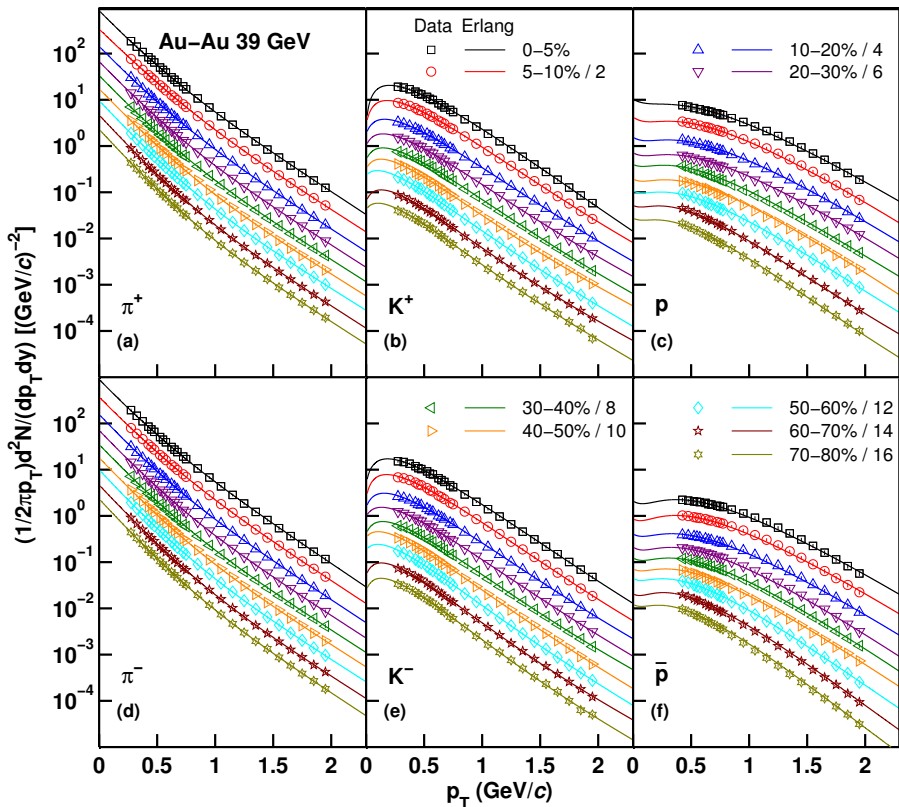

**Figure 5.** Same as Figure 1 for Au-Au collisions at $\sqrt{s_{NN}} = 39$ GeV. The experimental data are from reference [33].

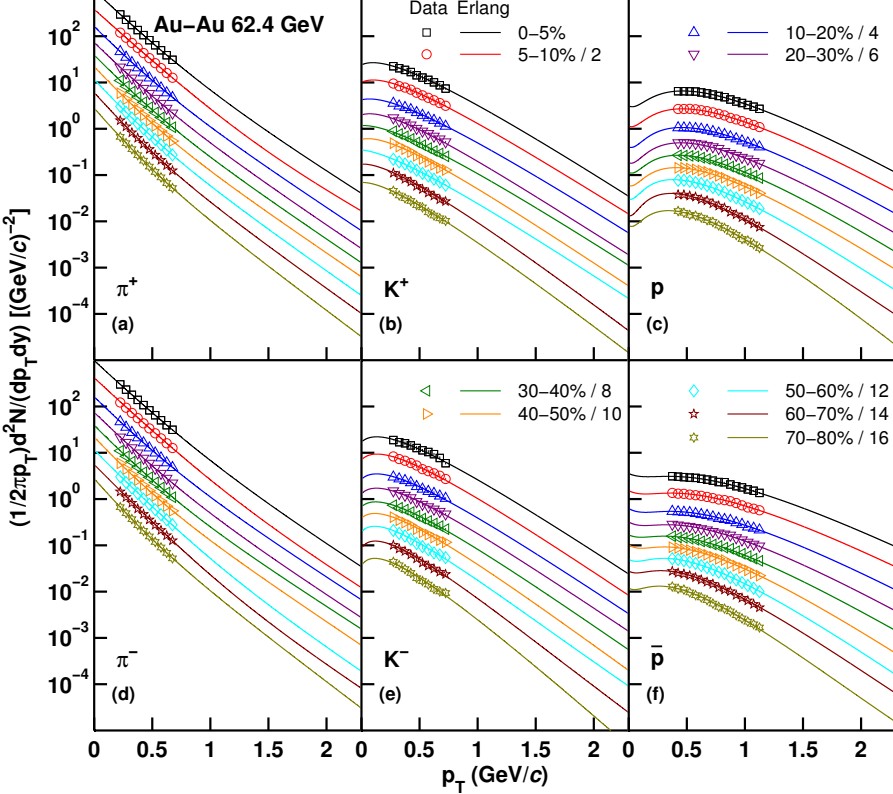

**Figure 6.** Same as Figure 1 for Au-Au collisions at $\sqrt{s_{NN}} = 62.4$ GeV. The experimental data were recorded by the STAR Collaboration [34].

**Table 1.** Values of free parameters, normalization constant, and $\chi^2/$dof corresponding to the two-component Erlang $p_T$ distribution for production in Au-Au collisions at $\sqrt{s_{NN}} = 7.7$ GeV for different centralities in Figure 1. ($m_1$ equals 2, 3, and 4 for $\pi^\pm$, $K^\pm$, and $p(\bar{p})$, respectively; $m_2$ equals 2 for all particles.)

| Figure | Particle | Centrality | $< p_{ti1} >$ (GeV/c) | $k_1$ | $< p_{ti2} >$ (GeV/c) | $N_0$ | $\chi^2/$dof |
|---|---|---|---|---|---|---|---|
| Figure 1a | $\pi^+$ | 0–5% | 0.172 ± 0.004 | 0.63 ± 0.06 | 0.233 ± 0.004 | 96.122 ± 3.364 | 13.970/20 |
| | | 5–10% | 0.148 ± 0.006 | 0.51 ± 0.03 | 0.225 ± 0.002 | 80.050 ± 2.642 | 3.319/20 |
| | | 10–20% | 0.150 ± 0.006 | 0.51 ± 0.04 | 0.224 ± 0.003 | 60.889 ± 2.192 | 1.896/20 |
| | | 20–30% | 0.150 ± 0.006 | 0.51 ± 0.03 | 0.219 ± 0.002 | 41.772 ± 1.378 | 3.579/20 |
| | | 30–40% | 0.147 ± 0.006 | 0.51 ± 0.03 | 0.214 ± 0.003 | 28.090 ± 0.955 | 5.519/20 |
| | | 40–50% | 0.135 ± 0.005 | 0.51 ± 0.03 | 0.207 ± 0.003 | 18.994 ± 0.646 | 3.120/19 |
| | | 50–60% | 0.133 ± 0.006 | 0.51 ± 0.03 | 0.198 ± 0.003 | 11.956 ± 0.383 | 8.153/19 |
| | | 60–70% | 0.148 ± 0.004 | 0.59 ± 0.05 | 0.195 ± 0.003 | 6.225 ± 0.212 | 9.046/18 |
| | | 70–80% | 0.151 ± 0.004 | 0.53 ± 0.07 | 0.176 ± 0.004 | 3.125 ± 0.103 | 3.702/15 |
| Figure 1d | $\pi^-$ | 0–5% | 0.149 ± 0.006 | 0.52 ± 0.03 | 0.219 ± 0.003 | 107.122 ± 3.642 | 9.297/20 |
| | | 5–10% | 0.147 ± 0.006 | 0.51 ± 0.03 | 0.216 ± 0.002 | 86.526 ± 3.028 | 16.388/20 |
| | | 10–20% | 0.141 ± 0.007 | 0.51 ± 0.03 | 0.219 ± 0.002 | 67.409 ± 2.494 | 3.500/20 |
| | | 20–30% | 0.142 ± 0.006 | 0.51 ± 0.03 | 0.215 ± 0.003 | 45.954 ± 1.608 | 5.489/20 |
| | | 30–40% | 0.147 ± 0.007 | 0.51 ± 0.03 | 0.209 ± 0.003 | 30.232 ± 1.028 | 7.770/20 |
| | | 40–50% | 0.137 ± 0.006 | 0.51 ± 0.03 | 0.203 ± 0.003 | 20.251 ± 0.689 | 9.656/18 |
| | | 50–60% | 0.132 ± 0.006 | 0.51 ± 0.03 | 0.195 ± 0.002 | 12.950 ± 0.414 | 7.004/17 |
| | | 60–70% | 0.161 ± 0.003 | 0.83 ± 0.05 | 0.200 ± 0.006 | 6.606 ± 0.231 | 9.005/17 |
| | | 70–80% | 0.146 ± 0.004 | 0.61 ± 0.05 | 0.185 ± 0.004 | 3.438 ± 0.131 | 2.791/15 |
| Figure 1b | $K^+$ | 0–5% | 0.197 ± 0.003 | 0.81 ± 0.16 | 0.260 ± 0.018 | 20.070 ± 0.682 | 9.050/17 |
| | | 5–10% | 0.193 ± 0.003 | 0.86 ± 0.14 | 0.270 ± 0.024 | 16.565 ± 0.514 | 6.988/19 |
| | | 10–20% | 0.191 ± 0.002 | 0.86 ± 0.14 | 0.255 ± 0.017 | 12.444 ± 0.373 | 11.722/19 |
| | | 20–30% | 0.186 ± 0.002 | 0.87 ± 0.13 | 0.244 ± 0.016 | 8.508 ± 0.264 | 6.350/19 |
| | | 30–40% | 0.178 ± 0.003 | 0.85 ± 0.15 | 0.252 ± 0.016 | 5.111 ± 0.164 | 13.359/18 |
| | | 40–50% | 0.173 ± 0.003 | 0.86 ± 0.14 | 0.255 ± 0.043 | 3.150 ± 0.101 | 7.742/17 |
| | | 50–60% | 0.168 ± 0.002 | 0.90 ± 0.10 | 0.231 ± 0.032 | 1.721 ± 0.053 | 5.944/16 |
| | | 60–70% | 0.161 ± 0.002 | 0.92 ± 0.08 | 0.202 ± 0.040 | 0.800 ± 0.034 | 8.731/15 |
| | | 70–80% | 0.156 ± 0.003 | 0.91 ± 0.09 | 0.230 ± 0.046 | 0.330 ± 0.013 | 13.035/12 |
| Figure 1e | $K^-$ | 0–5% | 0.185 ± 0.004 | 0.89 ± 0.11 | 0.258 ± 0.015 | 7.208 ± 0.252 | 18.217/17 |
| | | 5–10% | 0.182 ± 0.003 | 0.82 ± 0.16 | 0.247 ± 0.018 | 6.131 ± 0.196 | 10.748/17 |
| | | 10–20% | 0.182 ± 0.002 | 0.86 ± 0.14 | 0.226 ± 0.024 | 4.620 ± 0.143 | 4.469/17 |
| | | 20–30% | 0.174 ± 0.002 | 0.92 ± 0.08 | 0.195 ± 0.039 | 3.129 ± 0.113 | 4.821/17 |
| | | 30–40% | 0.170 ± 0.002 | 0.95 ± 0.05 | 0.204 ± 0.040 | 1.990 ± 0.070 | 6.867/17 |
| | | 40–50% | 0.162 ± 0.002 | 0.90 ± 0.10 | 0.220 ± 0.044 | 1.243 ± 0.044 | 8.169/14 |
| | | 50–60% | 0.155 ± 0.002 | 0.90 ± 0.10 | 0.195 ± 0.034 | 0.692 ± 0.027 | 12.526/15 |
| | | 60–70% | 0.152 ± 0.003 | 0.90 ± 0.10 | 0.218 ± 0.043 | 0.312 ± 0.011 | 12.456/13 |
| | | 70–80% | 0.146 ± 0.004 | 0.91 ± 0.09 | 0.180 ± 0.036 | 0.133 ± 0.006 | 27.764/10 |
| Figure 1c | $p$ | 0–5% | 0.215 ± 0.003 | 0.89 ± 0.08 | 0.270 ± 0.054 | 52.211 ± 2.193 | 4.927/23 |
| | | 5–10% | 0.211 ± 0.003 | 0.91 ± 0.08 | 0.265 ± 0.053 | 44.223 ± 1.946 | 3.353/23 |
| | | 10–20% | 0.201 ± 0.003 | 0.91 ± 0.09 | 0.265 ± 0.053 | 32.020 ± 1.473 | 9.462/23 |
| | | 20–30% | 0.200 ± 0.003 | 0.92 ± 0.08 | 0.250 ± 0.050 | 21.932 ± 0.855 | 2.729/23 |
| | | 30–40% | 0.192 ± 0.003 | 0.90 ± 0.09 | 0.250 ± 0.050 | 14.565 ± 0.612 | 4.083/22 |
| | | 40–50% | 0.183 ± 0.003 | 0.86 ± 0.09 | 0.260 ± 0.035 | 8.745 ± 0.350 | 4.582/22 |
| | | 50–60% | 0.174 ± 0.003 | 0.83 ± 0.14 | 0.260 ± 0.035 | 5.248 ± 0.210 | 18.492/21 |
| | | 60–70% | 0.164 ± 0.003 | 0.80 ± 0.16 | 0.260 ± 0.019 | 2.622 ± 0.105 | 13.162/22 |
| | | 70–80% | 0.154 ± 0.002 | 0.71 ± 0.16 | 0.200 ± 0.016 | 1.345 ± 0.059 | 12.753/15 |
| Figure 1f | $\bar{p}$ | 0–5% | 0.232 ± 0.011 | 0.81 ± 0.16 | 0.334 ± 0.066 | 0.392 ± 0.019 | 3.384/9 |
| | | 5–10% | 0.217 ± 0.009 | 0.87 ± 0.13 | 0.247 ± 0.049 | 0.338 ± 0.016 | 7.632/8 |
| | | 10–20% | 0.205 ± 0.009 | 0.82 ± 0.11 | 0.302 ± 0.060 | 0.257 ± 0.012 | 3.345/12 |
| | | 20–30% | 0.198 ± 0.009 | 0.80 ± 0.16 | 0.310 ± 0.062 | 0.202 ± 0.009 | 9.058/10 |
| | | 30–40% | 0.184 ± 0.008 | 0.79 ± 0.15 | 0.300 ± 0.060 | 0.140 ± 0.007 | 3.650/11 |
| | | 40–50% | 0.174 ± 0.006 | 0.81 ± 0.16 | 0.266 ± 0.053 | 0.099 ± 0.006 | 8.561/8 |
| | | 50–60% | 0.156 ± 0.007 | 0.79 ± 0.15 | 0.260 ± 0.052 | 0.058 ± 0.003 | 5.627/7 |
| | | 60–70% | 0.148 ± 0.007 | 0.83 ± 0.16 | 0.236 ± 0.047 | 0.031 ± 0.002 | 4.186/5 |
| | | 70–80% | 0.137 ± 0.011 | 0.82 ± 0.16 | 0.224 ± 0.044 | 0.017 ± 0.002 | 1.189/3 |

**Table 2.** Values of free parameters, normalization constant, and $\chi^2/\text{dof}$ corresponding to the two-component Erlang $p_T$ distribution for production in Au-Au collisions at $\sqrt{s_{NN}} = 11.5\,\text{GeV}$ for different centralities in Figure 2. ($m_1$ equals 2, 3, and 4 for $\pi^\pm$, $K^\pm$, and $p(\bar{p})$, respectively; $m_2$ equals 2 for all particles.)

| Figure | Particle | Centrality | $<p_{ti1}>$ (GeV/c) | $k_1$ | $<p_{ti2}>$ (GeV/c) | $N_0$ | $\chi^2/\text{dof}$ |
|---|---|---|---|---|---|---|---|
| Figure 2a | $\pi^+$ | 0–5% | $0.153 \pm 0.007$ | $0.52 \pm 0.04$ | $0.236 \pm 0.003$ | $125.208 \pm 4.633$ | 2.186/20 |
| | | 5–10% | $0.153 \pm 0.006$ | $0.51 \pm 0.04$ | $0.233 \pm 0.003$ | $98.692 \pm 3.257$ | 1.584/20 |
| | | 10–20% | $0.149 \pm 0.007$ | $0.51 \pm 0.03$ | $0.230 \pm 0.003$ | $76.333 \pm 3.053$ | 1.429/20 |
| | | 20–30% | $0.143 \pm 0.006$ | $0.53 \pm 0.04$ | $0.233 \pm 0.003$ | $52.743 \pm 1.846$ | 1.146/20 |
| | | 30–40% | $0.141 \pm 0.006$ | $0.53 \pm 0.04$ | $0.227 \pm 0.003$ | $36.192 \pm 1.231$ | 0.998/20 |
| | | 40–50% | $0.133 \pm 0.006$ | $0.51 \pm 0.03$ | $0.221 \pm 0.003$ | $23.473 \pm 0.845$ | 1.222/20 |
| | | 50–60% | $0.132 \pm 0.006$ | $0.51 \pm 0.03$ | $0.212 \pm 0.002$ | $14.263 \pm 0.571$ | 2.065/20 |
| | | 60–70% | $0.133 \pm 0.006$ | $0.51 \pm 0.03$ | $0.204 \pm 0.003$ | $8.159 \pm 0.310$ | 2.188/18 |
| | | 70–80% | $0.141 \pm 0.006$ | $0.52 \pm 0.04$ | $0.197 \pm 0.003$ | $4.135 \pm 0.141$ | 5.479/18 |
| Figure 2d | $\pi^-$ | 0–5% | $0.146 \pm 0.008$ | $0.51 \pm 0.04$ | $0.230 \pm 0.003$ | $135.170 \pm 5.812$ | 1.593/20 |
| | | 5–10% | $0.143 \pm 0.007$ | $0.51 \pm 0.04$ | $0.229 \pm 0.002$ | $107.425 \pm 4.082$ | 1.383/20 |
| | | 10–20% | $0.140 \pm 0.005$ | $0.53 \pm 0.04$ | $0.230 \pm 0.003$ | $83.065 \pm 2.658$ | 0.781/20 |
| | | 20–30% | $0.135 \pm 0.006$ | $0.51 \pm 0.04$ | $0.227 \pm 0.003$ | $56.923 \pm 1.992$ | 0.895/20 |
| | | 30–40% | $0.140 \pm 0.006$ | $0.51 \pm 0.03$ | $0.220 \pm 0.003$ | $38.112 \pm 1.334$ | 2.378/20 |
| | | 40–50% | $0.137 \pm 0.007$ | $0.51 \pm 0.03$ | $0.216 \pm 0.003$ | $24.354 \pm 0.950$ | 4.048/20 |
| | | 50–60% | $0.137 \pm 0.006$ | $0.51 \pm 0.03$ | $0.209 \pm 0.003$ | $14.725 \pm 0.560$ | 7.309/20 |
| | | 60–70% | $0.139 \pm 0.005$ | $0.51 \pm 0.03$ | $0.201 \pm 0.003$ | $8.447 \pm 0.287$ | 11.989/18 |
| | | 70–80% | $0.150 \pm 0.005$ | $0.51 \pm 0.04$ | $0.191 \pm 0.003$ | $4.250 \pm 0.153$ | 10.734/18 |
| Figure 2b | $K^+$ | 0–5% | $0.201 \pm 0.003$ | $0.88 \pm 0.12$ | $0.262 \pm 0.048$ | $24.436 \pm 0.733$ | 1.486/19 |
| | | 5–10% | $0.198 \pm 0.002$ | $0.94 \pm 0.06$ | $0.207 \pm 0.041$ | $19.832 \pm 0.714$ | 2.160/20 |
| | | 10–20% | $0.200 \pm 0.002$ | $0.82 \pm 0.06$ | $0.199 \pm 0.026$ | $14.781 \pm 0.473$ | 2.322/20 |
| | | 20–30% | $0.199 \pm 0.003$ | $0.76 \pm 0.06$ | $0.205 \pm 0.020$ | $9.726 \pm 0.350$ | 3.432/20 |
| | | 30–40% | $0.184 \pm 0.003$ | $0.79 \pm 0.05$ | $0.306 \pm 0.010$ | $5.985 \pm 0.180$ | 1.553/20 |
| | | 40–50% | $0.174 \pm 0.003$ | $0.59 \pm 0.09$ | $0.260 \pm 0.007$ | $3.861 \pm 0.131$ | 5.020/20 |
| | | 50–60% | $0.180 \pm 0.003$ | $0.54 \pm 0.10$ | $0.236 \pm 0.006$ | $2.000 \pm 0.060$ | 5.888/19 |
| | | 60–70% | $0.172 \pm 0.003$ | $0.65 \pm 0.13$ | $0.228 \pm 0.009$ | $0.997 \pm 0.033$ | 2.330/17 |
| | | 70–80% | $0.163 \pm 0.003$ | $0.78 \pm 0.15$ | $0.231 \pm 0.016$ | $0.464 \pm 0.015$ | 11.598/16 |
| Figure 2e | $K^-$ | 0–5% | $0.191 \pm 0.003$ | $0.93 \pm 0.07$ | $0.221 \pm 0.044$ | $12.017 \pm 0.385$ | 1.338/17 |
| | | 5–10% | $0.188 \pm 0.002$ | $0.94 \pm 0.06$ | $0.295 \pm 0.059$ | $9.851 \pm 0.325$ | 7.987/18 |
| | | 10–20% | $0.193 \pm 0.003$ | $0.83 \pm 0.16$ | $0.219 \pm 0.022$ | $7.509 \pm 0.225$ | 10.112/18 |
| | | 20–30% | $0.185 \pm 0.003$ | $0.82 \pm 0.16$ | $0.245 \pm 0.018$ | $4.915 \pm 0.152$ | 3.830/18 |
| | | 30–40% | $0.182 \pm 0.002$ | $0.88 \pm 0.11$ | $0.256 \pm 0.015$ | $3.046 \pm 0.091$ | 3.806/17 |
| | | 40–50% | $0.167 \pm 0.003$ | $0.82 \pm 0.06$ | $0.275 \pm 0.010$ | $1.974 \pm 0.069$ | 4.685/17 |
| | | 50–60% | $0.164 \pm 0.003$ | $0.92 \pm 0.04$ | $0.327 \pm 0.027$ | $1.031 \pm 0.032$ | 4.597/17 |
| | | 60–70% | $0.162 \pm 0.003$ | $0.92 \pm 0.08$ | $0.254 \pm 0.050$ | $0.517 \pm 0.019$ | 5.856/14 |
| | | 70–80% | $0.148 \pm 0.004$ | $0.91 \pm 0.06$ | $0.325 \pm 0.046$ | $0.246 \pm 0.010$ | 14.250/10 |
| Figure 2c | $p$ | 0–5% | $0.213 \pm 0.004$ | $0.88 \pm 0.08$ | $0.234 \pm 0.046$ | $42.924 \pm 1.803$ | 8.894/22 |
| | | 5–10% | $0.214 \pm 0.005$ | $0.89 \pm 0.10$ | $0.230 \pm 0.046$ | $34.265 \pm 1.508$ | 1.597/23 |
| | | 10–20% | $0.211 \pm 0.004$ | $0.88 \pm 0.08$ | $0.228 \pm 0.045$ | $25.603 \pm 1.203$ | 1.361/23 |
| | | 20–30% | $0.205 \pm 0.004$ | $0.85 \pm 0.12$ | $0.250 \pm 0.050$ | $17.849 \pm 0.696$ | 6.665/23 |
| | | 30–40% | $0.200 \pm 0.004$ | $0.80 \pm 0.16$ | $0.285 \pm 0.038$ | $11.424 \pm 0.434$ | 5.483/23 |
| | | 40–50% | $0.189 \pm 0.003$ | $0.80 \pm 0.16$ | $0.275 \pm 0.029$ | $7.038 \pm 0.282$ | 11.643/22 |
| | | 50–60% | $0.179 \pm 0.003$ | $0.70 \pm 0.14$ | $0.286 \pm 0.017$ | $4.076 \pm 0.167$ | 12.886/22 |
| | | 60–70% | $0.171 \pm 0.003$ | $0.62 \pm 0.14$ | $0.230 \pm 0.010$ | $2.208 \pm 0.097$ | 14.500/22 |
| | | 70–80% | $0.161 \pm 0.003$ | $0.69 \pm 0.13$ | $0.246 \pm 0.015$ | $0.995 \pm 0.040$ | 24.989/23 |
| Figure 2f | $\bar{p}$ | 0–5% | $0.216 \pm 0.003$ | $0.88 \pm 0.08$ | $0.234 \pm 0.046$ | $1.374 \pm 0.066$ | 23.483/17 |
| | | 5–10% | $0.209 \pm 0.005$ | $0.84 \pm 0.14$ | $0.273 \pm 0.054$ | $1.098 \pm 0.047$ | 21.419/17 |
| | | 10–20% | $0.203 \pm 0.004$ | $0.85 \pm 0.10$ | $0.253 \pm 0.050$ | $0.888 \pm 0.039$ | 24.352/17 |
| | | 20–30% | $0.196 \pm 0.004$ | $0.83 \pm 0.13$ | $0.265 \pm 0.045$ | $0.687 \pm 0.030$ | 8.201/17 |
| | | 30–40% | $0.189 \pm 0.004$ | $0.86 \pm 0.09$ | $0.229 \pm 0.045$ | $0.491 \pm 0.022$ | 4.604/17 |
| | | 40–50% | $0.176 \pm 0.003$ | $0.87 \pm 0.09$ | $0.214 \pm 0.042$ | $0.331 \pm 0.016$ | 5.954/14 |
| | | 50–60% | $0.172 \pm 0.004$ | $0.85 \pm 0.11$ | $0.228 \pm 0.045$ | $0.213 \pm 0.010$ | 3.503/13 |
| | | 60–70% | $0.155 \pm 0.004$ | $0.77 \pm 0.15$ | $0.261 \pm 0.034$ | $0.127 \pm 0.005$ | 3.407/8 |
| | | 70–80% | $0.147 \pm 0.008$ | $0.74 \pm 0.14$ | $0.256 \pm 0.051$ | $0.069 \pm 0.003$ | 7.499/8 |

**Table 3.** Values of free parameters, normalization constant, and $\chi^2/\mathrm{dof}$ corresponding to the two-component Erlang $p_T$ distribution for production in Au-Au collisions at $\sqrt{s_{NN}} = 19.6\,\mathrm{GeV}$ for different centralities in Figure 3. ($m_1$ equals 2, 3, and 4 for $\pi^{\pm}$, $K^{\pm}$, and $p(\bar{p})$, respectively; $m_2$ equals 2 for all particles.)

| Figure | Particle | Centrality | $< p_{ti1} >$ (GeV/c) | $k_1$ | $< p_{ti2} >$ (GeV/c) | $N_0$ | $\chi^2/\mathrm{dof}$ |
|---|---|---|---|---|---|---|---|
| Figure 3a | $\pi^+$ | 0–5% | $0.156 \pm 0.008$ | $0.57 \pm 0.05$ | $0.249 \pm 0.005$ | $165.077 \pm 8.089$ | 0.737/20 |
| | | 5–10% | $0.155 \pm 0.007$ | $0.61 \pm 0.04$ | $0.253 \pm 0.005$ | $136.482 \pm 5.732$ | 0.519/20 |
| | | 10–20% | $0.158 \pm 0.006$ | $0.63 \pm 0.04$ | $0.255 \pm 0.004$ | $103.435 \pm 4.137$ | 0.375/20 |
| | | 20–30% | $0.157 \pm 0.006$ | $0.66 \pm 0.03$ | $0.257 \pm 0.005$ | $70.526 \pm 2.962$ | 0.431/20 |
| | | 30–40% | $0.150 \pm 0.005$ | $0.64 \pm 0.04$ | $0.250 \pm 0.004$ | $48.064 \pm 1.730$ | 0.625/20 |
| | | 40–50% | $0.145 \pm 0.005$ | $0.63 \pm 0.04$ | $0.244 \pm 0.004$ | $30.629 \pm 1.072$ | 1.211/20 |
| | | 50–60% | $0.144 \pm 0.005$ | $0.65 \pm 0.03$ | $0.241 \pm 0.004$ | $18.732 \pm 0.693$ | 1.247/20 |
| | | 60–70% | $0.147 \pm 0.005$ | $0.68 \pm 0.03$ | $0.238 \pm 0.004$ | $10.253 \pm 0.390$ | 0.759/20 |
| | | 70–80% | $0.131 \pm 0.006$ | $0.53 \pm 0.03$ | $0.214 \pm 0.003$ | $5.523 \pm 0.232$ | 2.812/20 |
| Figure 3d | $\pi^-$ | 0–5% | $0.145 \pm 0.008$ | $0.56 \pm 0.04$ | $0.246 \pm 0.004$ | $176.077 \pm 8.452$ | 0.682/20 |
| | | 5–10% | $0.144 \pm 0.007$ | $0.58 \pm 0.04$ | $0.247 \pm 0.004$ | $145.211 \pm 6.244$ | 0.511/20 |
| | | 10–20% | $0.150 \pm 0.007$ | $0.61 \pm 0.04$ | $0.251 \pm 0.005$ | $108.850 \pm 5.007$ | 0.342/20 |
| | | 20–30% | $0.147 \pm 0.005$ | $0.62 \pm 0.04$ | $0.250 \pm 0.005$ | $74.545 \pm 2.684$ | 1.105/20 |
| | | 30–40% | $0.145 \pm 0.005$ | $0.63 \pm 0.04$ | $0.248 \pm 0.005$ | $50.278 \pm 1.860$ | 0.946/20 |
| | | 40–50% | $0.143 \pm 0.005$ | $0.63 \pm 0.03$ | $0.243 \pm 0.003$ | $31.803 \pm 1.177$ | 1.114/20 |
| | | 50–60% | $0.139 \pm 0.005$ | $0.61 \pm 0.03$ | $0.235 \pm 0.003$ | $19.461 \pm 0.681$ | 0.970/20 |
| | | 60–70% | $0.133 \pm 0.006$ | $0.56 \pm 0.03$ | $0.223 \pm 0.003$ | $10.814 \pm 0.411$ | 2.663/20 |
| | | 70–80% | $0.129 \pm 0.005$ | $0.51 \pm 0.03$ | $0.211 \pm 0.002$ | $5.709 \pm 0.183$ | 3.269/19 |
| Figure 3b | $K^+$ | 0–5% | $0.198 \pm 0.005$ | $0.67 \pm 0.13$ | $0.294 \pm 0.009$ | $29.706 \pm 0.891$ | 1.316/20 |
| | | 5–10% | $0.192 \pm 0.005$ | $0.68 \pm 0.07$ | $0.315 \pm 0.007$ | $24.335 \pm 0.973$ | 1.187/20 |
| | | 10–20% | $0.192 \pm 0.006$ | $0.57 \pm 0.10$ | $0.291 \pm 0.008$ | $18.218 \pm 0.619$ | 1.652/20 |
| | | 20–30% | $0.186 \pm 0.006$ | $0.53 \pm 0.09$ | $0.285 \pm 0.005$ | $12.546 \pm 0.452$ | 1.466/20 |
| | | 30–40% | $0.198 \pm 0.002$ | $0.69 \pm 0.13$ | $0.223 \pm 0.009$ | $8.233 \pm 0.255$ | 26.212/20 |
| | | 40–50% | $0.194 \pm 0.002$ | $0.63 \pm 0.12$ | $0.220 \pm 0.007$ | $4.992 \pm 0.155$ | 13.256/19 |
| | | 50–60% | $0.165 \pm 0.005$ | $0.51 \pm 0.05$ | $0.266 \pm 0.004$ | $2.790 \pm 0.089$ | 3.925/19 |
| | | 60–70% | $0.152 \pm 0.005$ | $0.51 \pm 0.04$ | $0.269 \pm 0.005$ | $1.417 \pm 0.054$ | 1.169/17 |
| | | 70–80% | $0.148 \pm 0.004$ | $0.57 \pm 0.05$ | $0.274 \pm 0.006$ | $0.680 \pm 0.023$ | 2.210/16 |
| Figure 3e | $K^-$ | 0–5% | $0.192 \pm 0.004$ | $0.64 \pm 0.07$ | $0.292 \pm 0.007$ | $18.620 \pm 0.596$ | 2.950/20 |
| | | 5–10% | $0.193 \pm 0.003$ | $0.70 \pm 0.09$ | $0.284 \pm 0.006$ | $15.498 \pm 0.527$ | 1.836/20 |
| | | 10–20% | $0.199 \pm 0.003$ | $0.65 \pm 0.13$ | $0.248 \pm 0.008$ | $11.714 \pm 0.375$ | 1.861/20 |
| | | 20–30% | $0.183 \pm 0.005$ | $0.56 \pm 0.08$ | $0.274 \pm 0.005$ | $8.148 \pm 0.253$ | 2.810/20 |
| | | 30–40% | $0.174 \pm 0.005$ | $0.51 \pm 0.09$ | $0.270 \pm 0.004$ | $5.358 \pm 0.166$ | 2.310/20 |
| | | 40–50% | $0.170 \pm 0.004$ | $0.58 \pm 0.06$ | $0.262 \pm 0.004$ | $3.299 \pm 0.102$ | 4.047/19 |
| | | 50–60% | $0.170 \pm 0.004$ | $0.51 \pm 0.10$ | $0.240 \pm 0.005$ | $1.920 \pm 0.060$ | 8.328/19 |
| | | 60–70% | $0.172 \pm 0.003$ | $0.51 \pm 0.10$ | $0.216 \pm 0.006$ | $0.991 \pm 0.035$ | 8.495/17 |
| | | 70–80% | $0.172 \pm 0.003$ | $0.65 \pm 0.05$ | $0.178 \pm 0.011$ | $0.472 \pm 0.015$ | 10.403/15 |
| Figure 3c | $p$ | 0–5% | $0.222 \pm 0.005$ | $0.79 \pm 0.11$ | $0.278 \pm 0.055$ | $34.690 \pm 1.353$ | 6.636/23 |
| | | 5–10% | $0.221 \pm 0.004$ | $0.84 \pm 0.08$ | $0.261 \pm 0.052$ | $28.720 \pm 1.120$ | 3.465/19 |
| | | 10–20% | $0.220 \pm 0.003$ | $0.85 \pm 0.15$ | $0.261 \pm 0.052$ | $22.471 \pm 0.921$ | 13.856/17 |
| | | 20–30% | $0.208 \pm 0.003$ | $0.84 \pm 0.07$ | $0.250 \pm 0.050$ | $14.238 \pm 0.541$ | 3.297/17 |
| | | 30–40% | $0.202 \pm 0.003$ | $0.81 \pm 0.10$ | $0.250 \pm 0.038$ | $9.105 \pm 0.355$ | 4.214/17 |
| | | 40–50% | $0.201 \pm 0.003$ | $0.82 \pm 0.14$ | $0.229 \pm 0.044$ | $5.738 \pm 0.258$ | 11.385/17 |
| | | 50–60% | $0.191 \pm 0.003$ | $0.81 \pm 0.16$ | $0.212 \pm 0.042$ | $3.252 \pm 0.140$ | 17.833/17 |
| | | 60–70% | $0.185 \pm 0.003$ | $0.82 \pm 0.09$ | $0.200 \pm 0.040$ | $1.684 \pm 0.082$ | 17.194/17 |
| | | 70–80% | $0.175 \pm 0.003$ | $0.78 \pm 0.09$ | $0.200 \pm 0.032$ | $0.818 \pm 0.038$ | 18.275/17 |
| Figure 3f | $\bar{p}$ | 0–5% | $0.222 \pm 0.004$ | $0.91 \pm 0.08$ | $0.247 \pm 0.049$ | $3.937 \pm 0.161$ | 11.586/16 |
| | | 5–10% | $0.215 \pm 0.003$ | $0.88 \pm 0.11$ | $0.290 \pm 0.058$ | $3.204 \pm 0.131$ | 15.482/16 |
| | | 10–20% | $0.213 \pm 0.004$ | $0.89 \pm 0.10$ | $0.240 \pm 0.048$ | $2.562 \pm 0.120$ | 12.446/18 |
| | | 20–30% | $0.209 \pm 0.004$ | $0.88 \pm 0.11$ | $0.240 \pm 0.048$ | $1.926 \pm 0.089$ | 2.372/18 |
| | | 30–40% | $0.199 \pm 0.003$ | $0.88 \pm 0.08$ | $0.249 \pm 0.049$ | $1.355 \pm 0.066$ | 1.118/19 |
| | | 40–50% | $0.190 \pm 0.004$ | $0.78 \pm 0.15$ | $0.313 \pm 0.029$ | $0.940 \pm 0.038$ | 3.909/19 |
| | | 50–60% | $0.177 \pm 0.003$ | $0.79 \pm 0.15$ | $0.300 \pm 0.018$ | $0.586 \pm 0.028$ | 1.950/19 |
| | | 60–70% | $0.167 \pm 0.003$ | $0.81 \pm 0.15$ | $0.268 \pm 0.023$ | $0.340 \pm 0.014$ | 7.299/17 |
| | | 70–80% | $0.155 \pm 0.003$ | $0.81 \pm 0.14$ | $0.271 \pm 0.033$ | $0.169 \pm 0.007$ | 4.949/16 |

**Table 4.** Values of free parameters, normalization constant, and $\chi^2/\text{dof}$ corresponding to the two-component Erlang $p_T$ distribution for production in Au-Au collisions at $\sqrt{s_{NN}} = 27\,\text{GeV}$ for different centralities in Figure 4. ($m_1$ equals 2, 3, and 4 for $\pi^\pm$, $K^\pm$, and $p(\bar{p})$, respectively; $m_2$ equals 2 for all particles.)

| Figure | Particle | Centrality | $< p_{ti1} >$ (GeV/c) | $k_1$ | $< p_{ti2} >$ (GeV/c) | $N_0$ | $\chi^2/\text{dof}$ |
|---|---|---|---|---|---|---|---|
| Figure 4a | $\pi^+$ | 0–5% | $0.152 \pm 0.007$ | $0.51 \pm 0.05$ | $0.249 \pm 0.003$ | $182.402 \pm 6.202$ | $2.449/20$ |
| | | 5–10% | $0.155 \pm 0.007$ | $0.60 \pm 0.04$ | $0.257 \pm 0.005$ | $153.577 \pm 6.911$ | $0.311/20$ |
| | | 10–20% | $0.161 \pm 0.007$ | $0.64 \pm 0.04$ | $0.263 \pm 0.005$ | $114.051 \pm 4.676$ | $0.277/20$ |
| | | 20–30% | $0.158 \pm 0.007$ | $0.65 \pm 0.04$ | $0.263 \pm 0.005$ | $78.449 \pm 3.060$ | $0.423/20$ |
| | | 30–40% | $0.155 \pm 0.005$ | $0.66 \pm 0.04$ | $0.261 \pm 0.005$ | $52.828 \pm 2.007$ | $0.615/20$ |
| | | 40–50% | $0.160 \pm 0.005$ | $0.72 \pm 0.03$ | $0.268 \pm 0.006$ | $32.638 \pm 1.338$ | $0.568/20$ |
| | | 50–60% | $0.161 \pm 0.005$ | $0.74 \pm 0.03$ | $0.267 \pm 0.005$ | $19.253 \pm 0.712$ | $0.489/20$ |
| | | 60–70% | $0.152 \pm 0.004$ | $0.71 \pm 0.03$ | $0.255 \pm 0.004$ | $11.041 \pm 0.353$ | $0.909/20$ |
| | | 70–80% | $0.160 \pm 0.004$ | $0.79 \pm 0.02$ | $0.264 \pm 0.004$ | $5.287 \pm 0.169$ | $1.377/20$ |
| Figure 4d | $\pi^-$ | 0–5% | $0.164 \pm 0.006$ | $0.66 \pm 0.04$ | $0.264 \pm 0.004$ | $186.402 \pm 6.710$ | $2.325/20$ |
| | | 5–10% | $0.149 \pm 0.007$ | $0.60 \pm 0.04$ | $0.255 \pm 0.005$ | $160.852 \pm 6.273$ | $0.452/20$ |
| | | 10–20% | $0.157 \pm 0.007$ | $0.63 \pm 0.04$ | $0.261 \pm 0.005$ | $117.063 \pm 4.800$ | $0.337/20$ |
| | | 20–30% | $0.154 \pm 0.006$ | $0.63 \pm 0.04$ | $0.259 \pm 0.005$ | $80.615 \pm 2.983$ | $0.418/20$ |
| | | 30–40% | $0.160 \pm 0.005$ | $0.69 \pm 0.03$ | $0.267 \pm 0.006$ | $52.364 \pm 1.676$ | $0.442/20$ |
| | | 40–50% | $0.155 \pm 0.006$ | $0.68 \pm 0.03$ | $0.261 \pm 0.005$ | $33.302 \pm 1.565$ | $0.349/20$ |
| | | 50–60% | $0.150 \pm 0.005$ | $0.69 \pm 0.03$ | $0.257 \pm 0.005$ | $20.431 \pm 0.797$ | $0.675/20$ |
| | | 60–70% | $0.152 \pm 0.006$ | $0.71 \pm 0.03$ | $0.254 \pm 0.005$ | $11.201 \pm 0.538$ | $0.358/20$ |
| | | 70–80% | $0.144 \pm 0.005$ | $0.64 \pm 0.03$ | $0.235 \pm 0.003$ | $5.647 \pm 0.209$ | $1.634/20$ |
| Figure 4b | $K^+$ | 0–5% | $0.205 \pm 0.004$ | $0.97 \pm 0.02$ | $0.575 \pm 0.115$ | $29.993 \pm 0.930$ | $1.988/20$ |
| | | 5–10% | $0.201 \pm 0.003$ | $0.94 \pm 0.02$ | $0.462 \pm 0.030$ | $24.959 \pm 0.799$ | $2.304/20$ |
| | | 10–20% | $0.199 \pm 0.003$ | $0.92 \pm 0.03$ | $0.430 \pm 0.028$ | $18.851 \pm 0.547$ | $2.702/20$ |
| | | 20–30% | $0.191 \pm 0.004$ | $0.77 \pm 0.05$ | $0.349 \pm 0.010$ | $12.830 \pm 0.398$ | $1.806/20$ |
| | | 30–40% | $0.186 \pm 0.004$ | $0.77 \pm 0.03$ | $0.348 \pm 0.006$ | $8.241 \pm 0.247$ | $2.330/20$ |
| | | 40–50% | $0.174 \pm 0.004$ | $0.60 \pm 0.05$ | $0.307 \pm 0.006$ | $5.274 \pm 0.185$ | $1.779/20$ |
| | | 50–60% | $0.171 \pm 0.005$ | $0.64 \pm 0.05$ | $0.305 \pm 0.006$ | $2.935 \pm 0.091$ | $2.899/20$ |
| | | 60–70% | $0.165 \pm 0.003$ | $0.72 \pm 0.03$ | $0.313 \pm 0.007$ | $1.512 \pm 0.045$ | $4.541/20$ |
| | | 70–80% | $0.152 \pm 0.005$ | $0.51 \pm 0.05$ | $0.276 \pm 0.004$ | $0.702 \pm 0.022$ | $4.851/20$ |
| Figure 4e | $K^-$ | 0–5% | $0.198 \pm 0.002$ | $0.97 \pm 0.01$ | $0.531 \pm 0.052$ | $21.872 \pm 0.634$ | $5.463/19$ |
| | | 5–10% | $0.192 \pm 0.003$ | $0.85 \pm 0.03$ | $0.357 \pm 0.012$ | $18.306 \pm 0.567$ | $2.210/20$ |
| | | 10–20% | $0.188 \pm 0.003$ | $0.82 \pm 0.04$ | $0.351 \pm 0.010$ | $14.220 \pm 0.427$ | $2.037/20$ |
| | | 20–30% | $0.185 \pm 0.003$ | $0.81 \pm 0.03$ | $0.347 \pm 0.008$ | $9.713 \pm 0.291$ | $2.793/20$ |
| | | 30–40% | $0.183 \pm 0.003$ | $0.83 \pm 0.03$ | $0.353 \pm 0.009$ | $6.140 \pm 0.196$ | $3.571/20$ |
| | | 40–50% | $0.172 \pm 0.004$ | $0.65 \pm 0.05$ | $0.302 \pm 0.005$ | $3.911 \pm 0.121$ | $2.514/20$ |
| | | 50–60% | $0.169 \pm 0.004$ | $0.70 \pm 0.04$ | $0.303 \pm 0.005$ | $2.185 \pm 0.076$ | $2.350/20$ |
| | | 60–70% | $0.166 \pm 0.003$ | $0.82 \pm 0.03$ | $0.329 \pm 0.009$ | $1.101 \pm 0.034$ | $4.562/20$ |
| | | 70–80% | $0.163 \pm 0.004$ | $0.72 \pm 0.05$ | $0.276 \pm 0.008$ | $0.512 \pm 0.016$ | $12.189/20$ |
| Figure 4c | $p$ | 0–5% | $0.226 \pm 0.005$ | $0.80 \pm 0.16$ | $0.284 \pm 0.056$ | $30.191 \pm 1.328$ | $8.619/17$ |
| | | 5–10% | $0.222 \pm 0.005$ | $0.82 \pm 0.10$ | $0.261 \pm 0.052$ | $26.024 \pm 1.093$ | $7.610/17$ |
| | | 10–20% | $0.222 \pm 0.004$ | $0.83 \pm 0.10$ | $0.261 \pm 0.052$ | $20.160 \pm 0.887$ | $4.736/17$ |
| | | 20–30% | $0.216 \pm 0.004$ | $0.78 \pm 0.15$ | $0.308 \pm 0.035$ | $13.750 \pm 0.577$ | $9.822/17$ |
| | | 30–40% | $0.206 \pm 0.004$ | $0.70 \pm 0.14$ | $0.345 \pm 0.022$ | $9.066 \pm 0.381$ | $13.078/17$ |
| | | 40–50% | $0.196 \pm 0.006$ | $0.65 \pm 0.13$ | $0.339 \pm 0.020$ | $5.287 \pm 0.248$ | $2.337/17$ |
| | | 50–60% | $0.185 \pm 0.004$ | $0.67 \pm 0.13$ | $0.318 \pm 0.018$ | $3.037 \pm 0.118$ | $6.823/17$ |
| | | 60–70% | $0.175 \pm 0.004$ | $0.64 \pm 0.12$ | $0.305 \pm 0.013$ | $1.725 \pm 0.069$ | $23.466/17$ |
| | | 70–80% | $0.160 \pm 0.004$ | $0.57 \pm 0.08$ | $0.293 \pm 0.008$ | $0.733 \pm 0.029$ | $5.857/17$ |
| Figure 4f | $\bar{p}$ | 0–5% | $0.228 \pm 0.006$ | $0.78 \pm 0.15$ | $0.343 \pm 0.059$ | $5.877 \pm 0.235$ | $12.404/16$ |
| | | 5–10% | $0.228 \pm 0.005$ | $0.86 \pm 0.10$ | $0.260 \pm 0.052$ | $4.869 \pm 0.234$ | $9.255/16$ |
| | | 10–20% | $0.222 \pm 0.004$ | $0.87 \pm 0.12$ | $0.240 \pm 0.048$ | $3.884 \pm 0.179$ | $8.533/16$ |
| | | 20–30% | $0.217 \pm 0.004$ | $0.85 \pm 0.14$ | $0.241 \pm 0.048$ | $2.900 \pm 0.122$ | $1.710/16$ |
| | | 30–40% | $0.210 \pm 0.003$ | $0.83 \pm 0.13$ | $0.249 \pm 0.047$ | $2.072 \pm 0.085$ | $1.695/16$ |
| | | 40–50% | $0.200 \pm 0.004$ | $0.74 \pm 0.14$ | $0.295 \pm 0.025$ | $1.399 \pm 0.062$ | $1.579/16$ |
| | | 50–60% | $0.185 \pm 0.004$ | $0.73 \pm 0.14$ | $0.300 \pm 0.017$ | $0.820 \pm 0.032$ | $2.232/16$ |
| | | 60–70% | $0.177 \pm 0.003$ | $0.76 \pm 0.15$ | $0.277 \pm 0.017$ | $0.482 \pm 0.021$ | $21.340/16$ |
| | | 70–80% | $0.157 \pm 0.003$ | $0.67 \pm 0.13$ | $0.278 \pm 0.012$ | $0.229 \pm 0.009$ | $4.569/14$ |

**Table 5.** Values of free parameters, normalization constant, and $\chi^2/$dof corresponding to the two-component Erlang $p_T$ distribution for production in Au-Au collisions at $\sqrt{s_{NN}} = 39$ GeV for different centralities in Figure 5. ($m_1$ equals 2, 3, and 4 for $\pi^\pm$, $K^\pm$, and $p(\bar{p})$, respectively; $m_2$ equals 2 for all particles.)

| Figure | Particle | Centrality | $< p_{ti1} >$ (GeV/c) | $k_1$ | $< p_{ti2} >$ (GeV/c) | $N_0$ | $\chi^2/$dof |
|---|---|---|---|---|---|---|---|
| Figure 5a | $\pi^+$ | 0–5% | $0.155 \pm 0.008$ | $0.54 \pm 0.05$ | $0.265 \pm 0.004$ | $185.159 \pm 7.406$ | $3.265/20$ |
| | | 5–10% | $0.166 \pm 0.007$ | $0.63 \pm 0.04$ | $0.274 \pm 0.006$ | $153.984 \pm 6.159$ | $0.386/20$ |
| | | 10–20% | $0.158 \pm 0.008$ | $0.61 \pm 0.04$ | $0.270 \pm 0.005$ | $121.765 \pm 6.332$ | $0.376/20$ |
| | | 20–30% | $0.157 \pm 0.008$ | $0.63 \pm 0.04$ | $0.272 \pm 0.005$ | $83.486 \pm 4.091$ | $0.484/20$ |
| | | 30–40% | $0.158 \pm 0.007$ | $0.66 \pm 0.04$ | $0.275 \pm 0.006$ | $54.946 \pm 2.692$ | $0.493/20$ |
| | | 40–50% | $0.160 \pm 0.006$ | $0.70 \pm 0.03$ | $0.279 \pm 0.006$ | $34.963 \pm 1.538$ | $0.577/20$ |
| | | 50–60% | $0.156 \pm 0.004$ | $0.72 \pm 0.03$ | $0.279 \pm 0.006$ | $21.974 \pm 0.769$ | $0.740/20$ |
| | | 60–70% | $0.154 \pm 0.005$ | $0.72 \pm 0.03$ | $0.273 \pm 0.005$ | $12.108 \pm 0.533$ | $0.446/20$ |
| | | 70–80% | $0.153 \pm 0.006$ | $0.73 \pm 0.03$ | $0.272 \pm 0.006$ | $6.668 \pm 0.287$ | $0.419/20$ |
| Figure 5d | $\pi^-$ | 0–5% | $0.153 \pm 0.010$ | $0.53 \pm 0.04$ | $0.258 \pm 0.004$ | $191.409 \pm 7.274$ | $1.089/20$ |
| | | 5–10% | $0.159 \pm 0.008$ | $0.60 \pm 0.04$ | $0.266 \pm 0.006$ | $159.491 \pm 7.975$ | $0.233/20$ |
| | | 10–20% | $0.152 \pm 0.008$ | $0.59 \pm 0.04$ | $0.264 \pm 0.005$ | $126.386 \pm 6.446$ | $0.306/20$ |
| | | 20–30% | $0.155 \pm 0.007$ | $0.63 \pm 0.04$ | $0.269 \pm 0.005$ | $85.965 \pm 4.298$ | $0.365/20$ |
| | | 30–40% | $0.156 \pm 0.007$ | $0.65 \pm 0.04$ | $0.271 \pm 0.006$ | $56.262 \pm 2.869$ | $0.355/20$ |
| | | 40–50% | $0.158 \pm 0.006$ | $0.69 \pm 0.03$ | $0.274 \pm 0.006$ | $35.958 \pm 1.654$ | $0.376/20$ |
| | | 50–60% | $0.155 \pm 0.006$ | $0.71 \pm 0.03$ | $0.273 \pm 0.006$ | $22.498 \pm 1.080$ | $0.483/20$ |
| | | 60–70% | $0.156 \pm 0.006$ | $0.73 \pm 0.03$ | $0.273 \pm 0.005$ | $12.258 \pm 0.576$ | $0.425/20$ |
| | | 70–80% | $0.153 \pm 0.006$ | $0.72 \pm 0.03$ | $0.266 \pm 0.006$ | $6.781 \pm 0.353$ | $0.341/20$ |
| Figure 5b | $K^+$ | 0–5% | $0.211 \pm 0.003$ | $0.94 \pm 0.06$ | $0.359 \pm 0.045$ | $31.219 \pm 1.030$ | $5.650/20$ |
| | | 5–10% | $0.202 \pm 0.004$ | $0.83 \pm 0.05$ | $0.369 \pm 0.014$ | $27.111 \pm 0.840$ | $1.564/20$ |
| | | 10–20% | $0.198 \pm 0.005$ | $0.73 \pm 0.06$ | $0.349 \pm 0.010$ | $20.074 \pm 0.642$ | $1.340/20$ |
| | | 20–30% | $0.191 \pm 0.006$ | $0.67 \pm 0.06$ | $0.345 \pm 0.009$ | $13.603 \pm 0.422$ | $2.281/20$ |
| | | 30–40% | $0.189 \pm 0.005$ | $0.68 \pm 0.05$ | $0.337 \pm 0.006$ | $8.778 \pm 0.255$ | $2.462/20$ |
| | | 40–50% | $0.174 \pm 0.006$ | $0.60 \pm 0.05$ | $0.335 \pm 0.008$ | $5.514 \pm 0.210$ | $1.220/20$ |
| | | 50–60% | $0.168 \pm 0.006$ | $0.51 \pm 0.06$ | $0.304 \pm 0.004$ | $3.273 \pm 0.098$ | $2.751/20$ |
| | | 60–70% | $0.170 \pm 0.004$ | $0.63 \pm 0.04$ | $0.330 \pm 0.006$ | $1.605 \pm 0.048$ | $2.878/20$ |
| | | 70–80% | $0.164 \pm 0.005$ | $0.53 \pm 0.04$ | $0.304 \pm 0.005$ | $0.841 \pm 0.027$ | $6.889/20$ |
| Figure 5e | $K^-$ | 0–5% | $0.206 \pm 0.003$ | $0.86 \pm 0.06$ | $0.352 \pm 0.020$ | $24.658 \pm 0.715$ | $5.446/20$ |
| | | 5–10% | $0.199 \pm 0.004$ | $0.83 \pm 0.05$ | $0.354 \pm 0.014$ | $21.186 \pm 0.678$ | $3.110/20$ |
| | | 10–20% | $0.195 \pm 0.005$ | $0.73 \pm 0.05$ | $0.339 \pm 0.009$ | $15.792 \pm 0.553$ | $2.063/20$ |
| | | 20–30% | $0.190 \pm 0.006$ | $0.65 \pm 0.07$ | $0.321 \pm 0.008$ | $10.783 \pm 0.356$ | $2.454/20$ |
| | | 30–40% | $0.185 \pm 0.005$ | $0.68 \pm 0.05$ | $0.333 \pm 0.006$ | $7.005 \pm 0.224$ | $1.989/20$ |
| | | 40–50% | $0.176 \pm 0.005$ | $0.56 \pm 0.05$ | $0.301 \pm 0.005$ | $4.478 \pm 0.134$ | $2.504/20$ |
| | | 50–60% | $0.169 \pm 0.005$ | $0.55 \pm 0.05$ | $0.294 \pm 0.005$ | $2.666 \pm 0.080$ | $2.100/20$ |
| | | 60–70% | $0.168 \pm 0.004$ | $0.66 \pm 0.04$ | $0.317 \pm 0.005$ | $1.316 \pm 0.039$ | $3.159/20$ |
| | | 70–80% | $0.166 \pm 0.004$ | $0.67 \pm 0.04$ | $0.307 \pm 0.007$ | $0.677 \pm 0.022$ | $6.523/20$ |
| Figure 5c | $p$ | 0–5% | $0.239 \pm 0.005$ | $0.79 \pm 0.09$ | $0.293 \pm 0.055$ | $26.115 \pm 1.097$ | $3.078/16$ |
| | | 5–10% | $0.229 \pm 0.004$ | $0.86 \pm 0.08$ | $0.240 \pm 0.048$ | $22.026 \pm 1.035$ | $4.637/16$ |
| | | 10–20% | $0.229 \pm 0.006$ | $0.82 \pm 0.10$ | $0.281 \pm 0.056$ | $17.136 \pm 0.788$ | $1.181/16$ |
| | | 20–30% | $0.226 \pm 0.006$ | $0.80 \pm 0.16$ | $0.298 \pm 0.059$ | $12.027 \pm 0.469$ | $5.461/16$ |
| | | 30–40% | $0.209 \pm 0.006$ | $0.69 \pm 0.13$ | $0.355 \pm 0.032$ | $8.191 \pm 0.360$ | $10.732/16$ |
| | | 40–50% | $0.209 \pm 0.004$ | $0.75 \pm 0.15$ | $0.318 \pm 0.032$ | $4.934 \pm 0.227$ | $13.569/16$ |
| | | 50–60% | $0.196 \pm 0.005$ | $0.67 \pm 0.13$ | $0.341 \pm 0.026$ | $2.839 \pm 0.128$ | $7.375/16$ |
| | | 60–70% | $0.178 \pm 0.004$ | $0.62 \pm 0.11$ | $0.342 \pm 0.015$ | $1.411 \pm 0.062$ | $2.427/16$ |
| | | 70–80% | $0.171 \pm 0.005$ | $0.61 \pm 0.10$ | $0.317 \pm 0.009$ | $0.717 \pm 0.030$ | $5.298/16$ |
| Figure 5f | $\bar{p}$ | 0–5% | $0.237 \pm 0.004$ | $0.88 \pm 0.08$ | $0.270 \pm 0.054$ | $8.086 \pm 0.380$ | $9.659/17$ |
| | | 5–10% | $0.231 \pm 0.004$ | $0.85 \pm 0.11$ | $0.310 \pm 0.062$ | $6.970 \pm 0.293$ | $9.818/17$ |
| | | 10–20% | $0.228 \pm 0.005$ | $0.82 \pm 0.12$ | $0.306 \pm 0.060$ | $5.318 \pm 0.261$ | $6.490/17$ |
| | | 20–30% | $0.218 \pm 0.004$ | $0.86 \pm 0.10$ | $0.255 \pm 0.051$ | $3.722 \pm 0.164$ | $8.283/17$ |
| | | 30–40% | $0.213 \pm 0.004$ | $0.83 \pm 0.13$ | $0.274 \pm 0.054$ | $2.756 \pm 0.168$ | $0.376/17$ |
| | | 40–50% | $0.204 \pm 0.004$ | $0.74 \pm 0.14$ | $0.325 \pm 0.029$ | $1.824 \pm 0.071$ | $3.573/17$ |
| | | 50–60% | $0.189 \pm 0.003$ | $0.72 \pm 0.14$ | $0.311 \pm 0.013$ | $1.152 \pm 0.045$ | $3.976/17$ |
| | | 60–70% | $0.182 \pm 0.003$ | $0.75 \pm 0.15$ | $0.296 \pm 0.013$ | $0.636 \pm 0.028$ | $10.249/17$ |
| | | 70–80% | $0.172 \pm 0.003$ | $0.71 \pm 0.14$ | $0.281 \pm 0.014$ | $0.335 \pm 0.014$ | $23.783/17$ |

**Table 6.** Values of free parameters, normalization constant, and $\chi^2/\text{dof}$ corresponding to the two-component Erlang $p_T$ distribution for production in Au-Au collisions at $\sqrt{s_{NN}} = 62.4\,\text{GeV}$ for different centralities in Figure 6. ($m_1$ equals 2, 3, and 4 for $\pi^{\pm}$, $K^{\pm}$, and $p(\bar{p})$, respectively; $m_2$ equals 2 for all particles.)

| Figure | Particle | Centrality | $<p_{ti1}>$ (GeV/c) | $k_1$ | $<p_{ti2}>$ (GeV/c) | $N_0$ | $\chi^2/\text{dof}$ |
|---|---|---|---|---|---|---|---|
| Figure 6a | $\pi^+$ | 0–5% | 0.172 ± 0.003 | 0.65 ± 0.05 | 0.274 ± 0.012 | 232.461 ± 1.860 | 0.261/4 |
| | | 5–10% | 0.188 ± 0.003 | 0.85 ± 0.04 | 0.314 ± 0.022 | 187.816 ± 1.315 | 0.361/4 |
| | | 10–20% | 0.156 ± 0.003 | 0.51 ± 0.03 | 0.261 ± 0.009 | 146.014 ± 1.168 | 0.344/4 |
| | | 20–30% | 0.158 ± 0.002 | 0.51 ± 0.03 | 0.257 ± 0.008 | 99.959 ± 0.800 | 0.468/4 |
| | | 30–40% | 0.153 ± 0.003 | 0.51 ± 0.03 | 0.256 ± 0.009 | 67.869 ± 0.611 | 0.406/4 |
| | | 40–50% | 0.147 ± 0.002 | 0.51 ± 0.03 | 0.254 ± 0.008 | 44.560 ± 0.356 | 0.563/4 |
| | | 50–60% | 0.145 ± 0.002 | 0.51 ± 0.02 | 0.245 ± 0.008 | 27.279 ± 0.164 | 1.851/4 |
| | | 60–70% | 0.141 ± 0.002 | 0.53 ± 0.02 | 0.241 ± 0.011 | 15.409 ± 0.092 | 0.508/4 |
| | | 70–80% | 0.136 ± 0.002 | 0.51 ± 0.02 | 0.230 ± 0.008 | 7.645 ± 0.054 | 1.175/4 |
| Figure 6d | $\pi^-$ | 0–5% | 0.175 ± 0.002 | 0.67 ± 0.05 | 0.269 ± 0.008 | 234.954 ± 1.410 | 0.825/4 |
| | | 5–10% | 0.175 ± 0.003 | 0.73 ± 0.05 | 0.298 ± 0.026 | 193.787 ± 2.519 | 1.483/4 |
| | | 10–20% | 0.161 ± 0.003 | 0.56 ± 0.04 | 0.267 ± 0.009 | 147.565 ± 1.328 | 0.072/4 |
| | | 20–30% | 0.164 ± 0.003 | 0.59 ± 0.04 | 0.265 ± 0.011 | 101.980 ± 0.918 | 0.091/4 |
| | | 30–40% | 0.159 ± 0.003 | 0.58 ± 0.04 | 0.269 ± 0.012 | 68.737 ± 0.619 | 0.120/4 |
| | | 40–50% | 0.148 ± 0.002 | 0.51 ± 0.04 | 0.257 ± 0.007 | 44.980 ± 0.270 | 0.867/4 |
| | | 50–60% | 0.160 ± 0.002 | 0.69 ± 0.06 | 0.297 ± 0.018 | 27.748 ± 0.361 | 0.148/4 |
| | | 60–70% | 0.157 ± 0.002 | 0.71 ± 0.04 | 0.303 ± 0.021 | 15.342 ± 0.199 | 0.412/4 |
| | | 70–80% | 0.138 ± 0.002 | 0.51 ± 0.02 | 0.228 ± 0.008 | 7.692 ± 0.046 | 0.572/4 |
| Figure 6b | $K^+$ | 0–5% | 0.246 ± 0.004 | 0.75 ± 0.04 | 0.256 ± 0.015 | 39.598 ± 0.752 | 0.299/4 |
| | | 5–10% | 0.236 ± 0.004 | 0.63 ± 0.10 | 0.310 ± 0.010 | 32.688 ± 0.490 | 1.425/4 |
| | | 10–20% | 0.248 ± 0.003 | 0.69 ± 0.06 | 0.269 ± 0.009 | 24.648 ± 0.320 | 1.034/4 |
| | | 20–30% | 0.226 ± 0.008 | 0.57 ± 0.11 | 0.302 ± 0.020 | 15.892 ± 0.445 | 5.988/4 |
| | | 30–40% | 0.247 ± 0.008 | 0.57 ± 0.10 | 0.252 ± 0.019 | 11.179 ± 0.291 | 0.500/4 |
| | | 40–50% | 0.224 ± 0.005 | 0.51 ± 0.11 | 0.266 ± 0.013 | 7.046 ± 0.127 | 2.632/4 |
| | | 50–60% | 0.234 ± 0.006 | 0.51 ± 0.07 | 0.275 ± 0.010 | 4.065 ± 0.081 | 0.345/4 |
| | | 60–70% | 0.217 ± 0.005 | 0.62 ± 0.12 | 0.231 ± 0.013 | 2.147 ± 0.045 | 3.387/4 |
| | | 70–80% | 0.213 ± 0.011 | 0.51 ± 0.12 | 0.233 ± 0.020 | 0.934 ± 0.031 | 5.179/4 |
| Figure 6e | $K^-$ | 0–5% | 0.235 ± 0.021 | 0.81 ± 0.14 | 0.281 ± 0.057 | 33.071 ± 1.885 | 7.217/4 |
| | | 5–10% | 0.226 ± 0.004 | 0.73 ± 0.14 | 0.288 ± 0.017 | 26.889 ± 0.511 | 3.348/4 |
| | | 10–20% | 0.228 ± 0.003 | 0.72 ± 0.12 | 0.291 ± 0.010 | 20.235 ± 0.243 | 3.616/4 |
| | | 20–30% | 0.222 ± 0.003 | 0.71 ± 0.14 | 0.292 ± 0.010 | 14.319 ± 0.172 | 3.191/4 |
| | | 30–40% | 0.218 ± 0.004 | 0.71 ± 0.09 | 0.279 ± 0.015 | 9.119 ± 0.173 | 2.298/4 |
| | | 40–50% | 0.208 ± 0.003 | 0.71 ± 0.14 | 0.265 ± 0.010 | 5.810 ± 0.070 | 5.979/4 |
| | | 50–60% | 0.195 ± 0.004 | 0.66 ± 0.13 | 0.282 ± 0.017 | 3.421 ± 0.072 | 4.352/4 |
| | | 60–70% | 0.187 ± 0.003 | 0.71 ± 0.14 | 0.261 ± 0.012 | 1.777 ± 0.023 | 5.638/4 |
| | | 70–80% | 0.178 ± 0.012 | 0.78 ± 0.16 | 0.194 ± 0.040 | 0.788 ± 0.037 | 8.929/4 |
| Figure 6c | $p$ | 0–5% | 0.256 ± 0.002 | 0.92 ± 0.04 | 0.336 ± 0.031 | 28.857 ± 0.231 | 4.730/9 |
| | | 5–10% | 0.253 ± 0.001 | 0.91 ± 0.03 | 0.382 ± 0.016 | 23.302 ± 0.140 | 7.145/9 |
| | | 10–20% | 0.247 ± 0.001 | 0.93 ± 0.04 | 0.347 ± 0.027 | 17.611 ± 0.123 | 6.489/9 |
| | | 20–30% | 0.237 ± 0.002 | 0.96 ± 0.02 | 0.315 ± 0.045 | 11.620 ± 0.116 | 20.228/9 |
| | | 30–40% | 0.229 ± 0.001 | 0.98 ± 0.02 | 0.273 ± 0.038 | 7.729 ± 0.046 | 6.932/9 |
| | | 40–50% | 0.221 ± 0.001 | 0.96 ± 0.04 | 0.330 ± 0.022 | 4.845 ± 0.029 | 6.636/9 |
| | | 50–60% | 0.209 ± 0.001 | 0.95 ± 0.05 | 0.380 ± 0.029 | 2.854 ± 0.017 | 13.302/9 |
| | | 60–70% | 0.197 ± 0.002 | 0.97 ± 0.03 | 0.360 ± 0.047 | 1.465 ± 0.010 | 13.381/9 |
| | | 70–80% | 0.188 ± 0.002 | 0.95 ± 0.04 | 0.207 ± 0.043 | 0.647 ± 0.006 | 20.066/9 |
| Figure 6f | $\bar{p}$ | 0–5% | 0.289 ± 0.002 | 0.77 ± 0.01 | 0.395 ± 0.007 | 15.211 ± 0.122 | 10.907/10 |
| | | 5–10% | 0.285 ± 0.002 | 0.80 ± 0.03 | 0.346 ± 0.011 | 12.675 ± 0.139 | 9.012/10 |
| | | 10–20% | 0.269 ± 0.002 | 0.84 ± 0.01 | 0.312 ± 0.013 | 9.551 ± 0.105 | 13.318/10 |
| | | 20–30% | 0.255 ± 0.002 | 0.82 ± 0.02 | 0.312 ± 0.018 | 6.532 ± 0.046 | 13.768/10 |
| | | 30–40% | 0.240 ± 0.002 | 0.77 ± 0.04 | 0.349 ± 0.013 | 4.335 ± 0.030 | 12.088/10 |
| | | 40–50% | 0.227 ± 0.002 | 0.81 ± 0.02 | 0.264 ± 0.020 | 2.811 ± 0.034 | 5.505/10 |
| | | 50–60% | 0.208 ± 0.002 | 0.79 ± 0.05 | 0.307 ± 0.011 | 1.688 ± 0.014 | 6.896/10 |
| | | 60–70% | 0.201 ± 0.002 | 0.67 ± 0.05 | 0.317 ± 0.021 | 0.963 ± 0.009 | 8.320/10 |
| | | 70–80% | 0.179 ± 0.003 | 0.67 ± 0.06 | 0.318 ± 0.034 | 0.433 ± 0.005 | 21.387/10 |

According to the extracted normalization constants from the above comparisons, the yield ratios of negative to positive particles, $k_\pi$, $k_K$, and $k_p$, versus collision energy and centrality are obtained. The three types of yield ratios show regular trends with an increase in collision energy and centrality. In order to see the dependencies of the three yield ratios on centrality, Figures 7–9, respectively, show the change trends of the three yield ratios of $k_\pi$, $k_K$, and $k_p$ with different centralities at different energies. As can be seen, $k_\pi$ varies by approximately 1.05 and decreases with increase in energy, but does not show a visible dependence on centrality. $k_K$ varies between 0.35 and 0.85, and increases obviously with increase of energy. At some energies (7.7, 11.5, 19.6 and 39 GeV), $k_K$ increases with increase in centrality class, but at these energies of 27 and 62.4 GeV, $k_K$ do not show an evident dependence on centrality. $k_p$ varies between 0.007 and 0.7, and prominently increases with the increase in energy. Unlike $k_\pi$ and $k_K$, $k_p$ obviously increases with increase in centrality class at all energies, which means that $k_p$ shows an obvious dependence on centrality. Overall, the dependence of $k_p$ on centrality is higher than that of $k_K$, and the dependence of $k_K$ on centrality is higher than that of $k_\pi$, which indicates that the correlation between the generation mechanism of $p$ ($\bar{p}$) and centrality is relatively the highest, followed by $K^\pm$, and $\pi^\pm$ is the weakest. In addition, it is not difficult to notice that with the increase in energy, the values of $k_K$ and $k_p$ are both less than 1 and gradually increase (most cases), while that of $k_\pi$ are almost equal to 1, which indicates that the generation mechanisms of these particles are closely related to the collision energy, and the generation mechanism of $p$ and $\bar{p}$ is similar to $K^\pm$, but different from $\pi^\pm$.

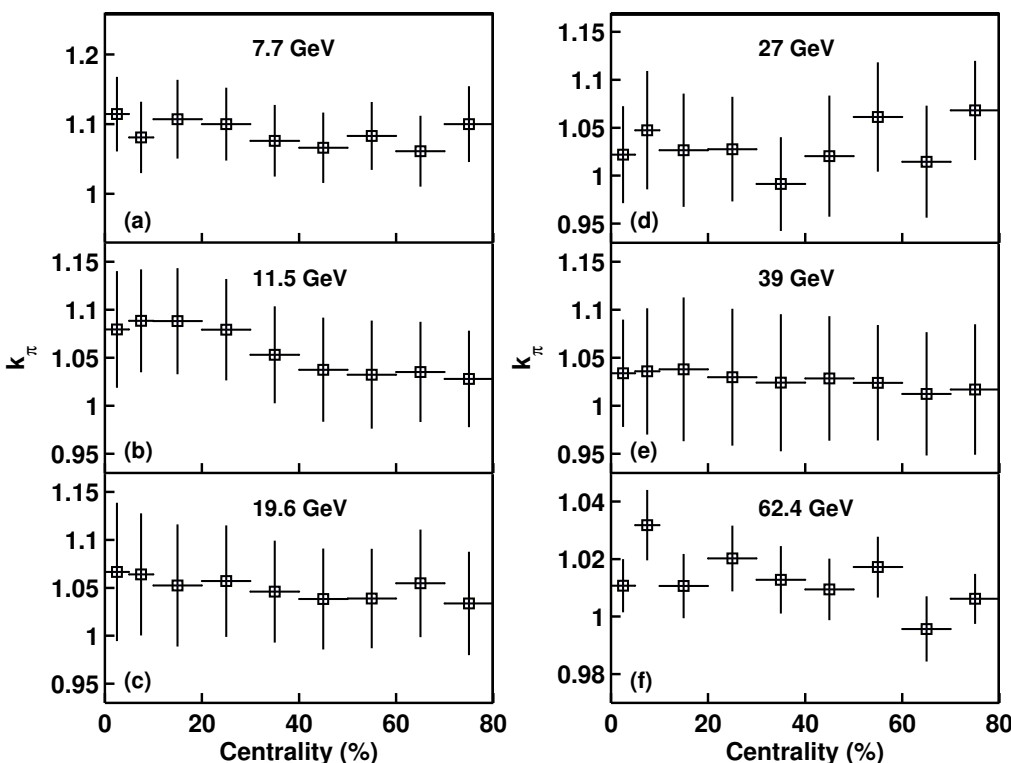

**Figure 7.** Centrality-dependent $k_\pi$ at different energies of (**a**) 7.7; (**b**) 11.5; (**c**) 19.6; (**d**) 27; (**e**) 39; and (**f**) 62.4 GeV.

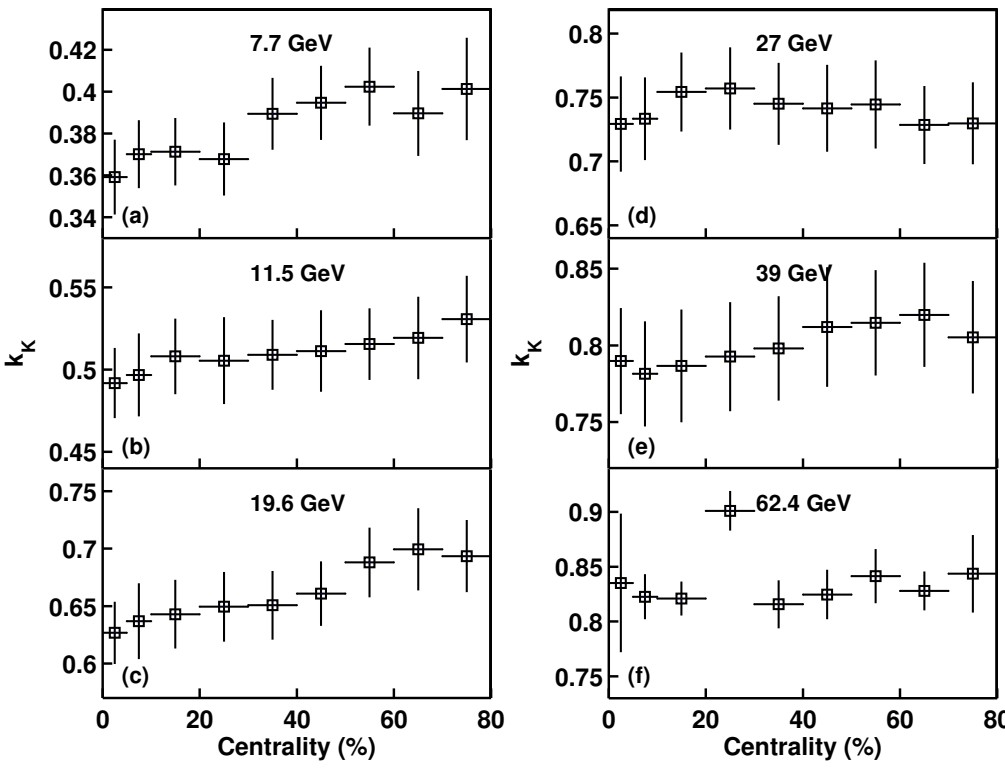

**Figure 8.** Same as Figure 7 for $k_K$.

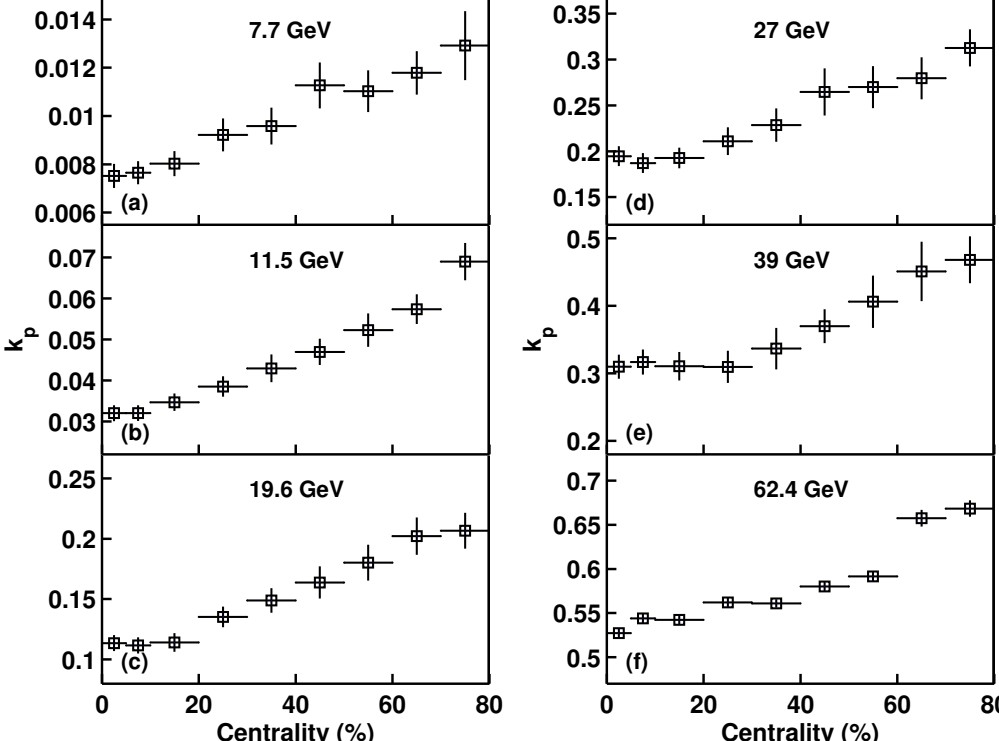

**Figure 9.** Same as Figure 7 for $k_p$.

In Figures 7–9, one can see that the three types of yield ratios obviously depend on the collision energy, and we find that the logarithms of the three types of yield ratios, $\ln(k_\pi)$,

$\ln(k_K)$, and $\ln(k_p)$, show a distinct linear dependence on $1/\sqrt{s_{NN}}$, a linear relationship which can be expressed as

$$\ln(k_{ij}) = A_{ij}/\sqrt{s_{NN}} + B_{ij}, \tag{11}$$

where $i$ represents $\pi$, $K$, or $p$, $j$ represent different centrality classes, and $A_{ij}$ and $B_{ij}$ are fitting parameters. Figure 10 shows the $1/\sqrt{s_{NN}}$-dependent (a) $\ln(k_\pi)$, (b) $\ln(k_K)$, and (c) $\ln(k_p)$ for different centralities. The fitting lines are the results calculated by the least squares method. The values of calculated parameters ($A_{ij}$ and $B_{ij}$) and $\chi^2/\mathrm{dof}$ are listed in Table 7. It is not hard to see that, the values of intercept $B_{ij}$ are asymptotically zero, which means the limiting values of the yield ratios are one at very high energy. For the same particle, the slope $A_{ij}$ does not change much with the increase in centrality, especially for $\pi$. To see clearly the dependences of the linear relationships on centrality, the results for different centrality classes are added by appropriate factors shown in different panels of Figure 10.

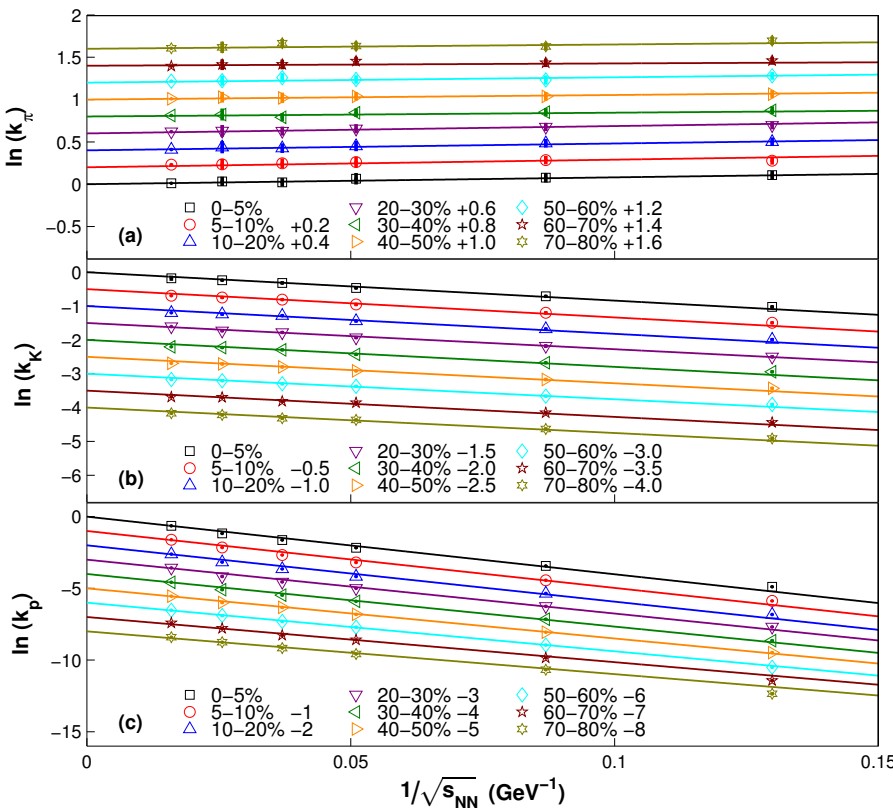

**Figure 10.** Energy-dependent (**a**) $\ln(k_\pi)$, (**b**) $\ln(k_K)$, and (**c**) $\ln(k_p)$ for different centralities. The fitting lines are the results calculated by least squares method. For clarity, the results for different centralities are added by appropriate factors shown in different panels.

As can be seen, with the increase in $\sqrt{s_{NN}}$, $\ln(k_K)$ and $\ln(k_p)$ increase, while $\ln(k_\pi)$ decreases, which implies that the generation mechanism of $K$ is similar to $p$, but is different from $\pi$. The differences of the cross-section of absorption, the content of primary proton in nuclei and so on can result in the differences of the yields of these particles. From Table 7, one can see that the centrality-dependent $A_\pi$ varies roughly between 0.48 and 0.89, and does not show an obvious change trend with centrality. $A_K$ and $A_p$ vary roughly from $-7.52$ to $-6.63$ and from $-38.86$ to $-33.42$, respectively, and an overall decrease with centrality, and the increase in $A_p$ is relatively prominent. These indicate that although the dependence of the energy-dependent yield ratio of $p$ on centrality is higher than that of

$K$, and that of $K$ is higher than that of $\pi$, the dependencies of the three energy-dependent yield ratios on centrality are not evident.

**Table 7.** Values of free parameters and $\chi^2/\text{dof}$ corresponding to the fitting lines in Figure 10.

| Figure | Particle | Centrality | $A_{ij}$ | $B_{ij}$ | $\chi^2/\text{dof}$ |
|--------|----------|------------|----------|----------|------------|
| | | 0–5% | $0.886 \pm 0.399$ | $-0.003 \pm 0.010$ | 0.209/3 |
| | | 5–10% | $0.507 \pm 0.376$ | $0.024 \pm 0.013$ | 0.229/3 |
| | | 10–20% | $0.880 \pm 0.459$ | $-0.003 \pm 0.012$ | 0.171/3 |
| | | 20–30% | $0.703 \pm 0.403$ | $0.009 \pm 0.012$ | 0.086/3 |
| Figure 10a | $\pi$ | 30–40% | $0.544 \pm 0.370$ | $0.003 \pm 0.013$ | 0.516/3 |
| | | 40–50% | $0.482 \pm 0.370$ | $0.002 \pm 0.011$ | 0.114/3 |
| | | 50–60% | $0.500 \pm 0.379$ | $0.010 \pm 0.012$ | 0.511/3 |
| | | 60–70% | $0.613 \pm 0.426$ | $-0.013 \pm 0.013$ | 0.507/3 |
| | | 70–80% | $0.692 \pm 0.331$ | $-0.004 \pm 0.010$ | 1.231/3 |
| | | 0–5% | $-7.522 \pm 0.444$ | $-0.055 \pm 0.032$ | 0.656/3 |
| | | 5–10% | $-7.086 \pm 0.349$ | $-0.075 \pm 0.028$ | 0.622/3 |
| | | 10–20% | $-6.956 \pm 0.424$ | $-0.076 \pm 0.030$ | 2.076/3 |
| | | 20–30% | $-7.922 \pm 0.501$ | $0.011 \pm 0.036$ | 1.985/3 |
| Figure 10b | K | 30–40% | $-6.667 \pm 0.313$ | $-0.081 \pm 0.027$ | 1.503/3 |
| | | 40–50% | $-6.641 \pm 0.324$ | $-0.073 \pm 0.026$ | 1.162/3 |
| | | 50–60% | $-6.688 \pm 0.323$ | $-0.053 \pm 0.023$ | 1.122/3 |
| | | 60–70% | $-6.637 \pm 0.499$ | $-0.068 \pm 0.040$ | 2.414/3 |
| | | 70–80% | $-6.570 \pm 0.496$ | $-0.056 \pm 0.029$ | 0.534/3 |
| | | 0–5% | $-38.697 \pm 1.583$ | $-0.031 \pm 0.138$ | 32.951/3 |
| | | 5–10% | $-38.855 \pm 3.000$ | $-0.002 \pm 0.200$ | 46.465/3 |
| | | 10–20% | $-38.147 \pm 1.051$ | $-0.014 \pm 0.092$ | 37.401/3 |
| | | 20–30% | $-37.168 \pm 0.600$ | $0.010 \pm 0.029$ | 23.505/3 |
| Figure 10c | $p$ | 30–40% | $-36.214 \pm 0.725$ | $-0.001 \pm 0.012$ | 7.260/3 |
| | | 40–50% | $-35.067 \pm 0.523$ | $0.014 \pm 0.014$ | 3.675/3 |
| | | 50–60% | $-34.785 \pm 0.479$ | $0.032 \pm 0.010$ | 1.071/3 |
| | | 60–70% | $-35.004 \pm 0.530$ | $0.140 \pm 0.016$ | 3.368/3 |
| | | 70–80% | $-33.417 \pm 0.978$ | $0.131 \pm 0.016$ | 4.916/3 |

Based on the extracted yield ratios and Equations (9) and (10), the energy- and centrality-dependent light hadron chemical potentials, $\mu_\pi$, $\mu_K$, and $\mu_p$, of $\pi$, $K$, and $p$, and quark chemical potentials, $\mu_u$, $\mu_d$, and $\mu_s$, of $u$, $d$, and $s$ quarks, respectively, were obtained and are shown in Figure 11. The different symbols denote the calculated results of different centrality classes. The curves are the derivative results according to Equation (11) corresponding to the fitted lines in Figure 10. As can be seen, in the energy range from 7.7 to 62.4 GeV, $\mu_\pi$ increases, and $\mu_K$, $\mu_p$, $\mu_u$, $\mu_d$, and $\mu_s$ obviously decrease with the increase in $\sqrt{s_{NN}}$. From the trends of the curves, the limiting values of the six types of chemical potentials are asymptotically zero at very high energy. The differences between chemical potentials of particles with different centralities are relatively large in the low energy region, and as the energy increases the differences gradually decrease, and finally tend to be zero at very high energy. These results are consistent with the results obtained in the references [41–43]. $\mu_p$ obtained in this work is smaller than $\mu_B$ in reference [43]. For an energy range from 7.7 to 39 GeV, the relative difference is mainly within 10%. However, the relative difference increases to approximately 20% at 62.4 GeV. In addition, at the same energy, $\mu_K$ is larger than $|\mu_\pi|$ but less than $\mu_p$, and $\mu_u$ is almost as large as $\mu_d$ but larger than $\mu_s$ due to the differences in the different particle masses.

It is not hard to notice that $\mu_\pi < 0$, while $\mu_K(\mu_p, \mu_u, \mu_d, \mu_s) > 0$. This is caused by $k_\pi > 1$, while $k_K(k_p) < 1$. When the energy increases to a very high value, all chemical potentials of light hadrons and quarks approach zero, when the partonic interactions possibly play a dominant role, the mean-free-path of particles becomes large, and the collision system possibly changes completely from the hadron-dominant state to the quark-dominant state.

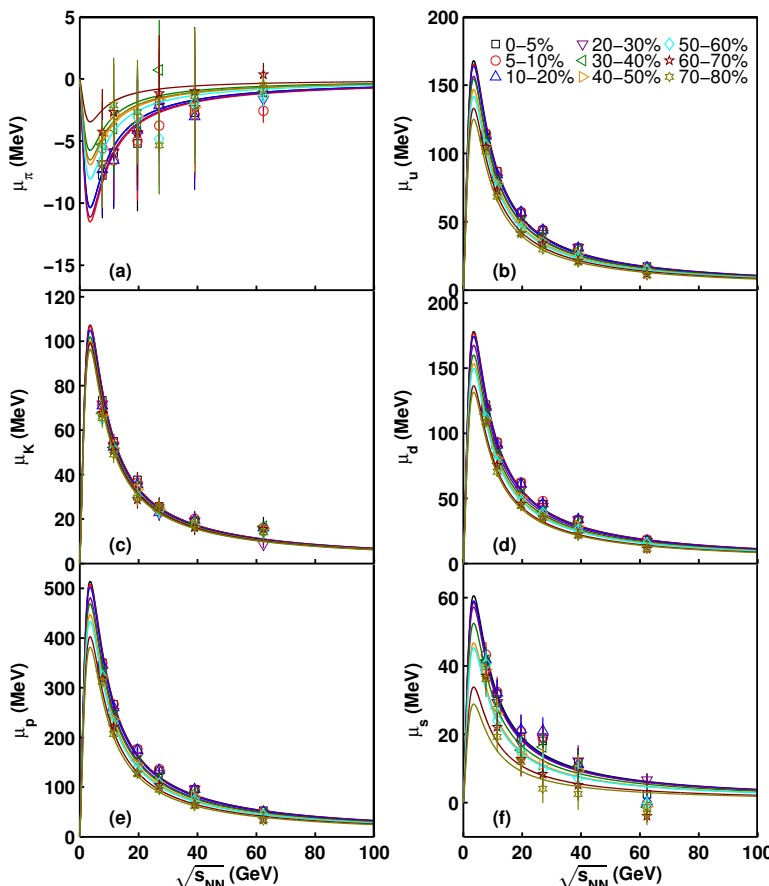

**Figure 11.** Dependencies of centrality-dependent hadron chemical potentials: (**a**) $\mu_\pi$; (**b**) $\mu_K$; and (**c**) $\mu_p$ and quark chemical potentials; (**d**) $\mu_u$; (**e**) $\mu_d$; and (**f**) $\mu_s$, on energy $\sqrt{s_{NN}}$. The different symbols denote the calculated results in different centrality classes according to the extracted yield ratios and Equations (9) and (10), and the curves are the derivative results based on the linear fits in Figure 10.

In Figure 11, as in our previous work [21], the derived curves of particle chemical potentials from the linear fits of the energy-dependent yield ratios in Figure 10 simultaneously show the maximum (the absolute magnitude for $\pi$) at 3.526 GeV, which is not observed from the linear fits. The energy at the maximum can be obtained according to the calculation method in reference [21]. According to Equations (7) and (9)–(11), we can obtain the chemical potentials ($\mu_{kj}$) of hadrons and quarks for all centrality classes in terms of $\sqrt{s_{NN}}$. Considering that all the values of intercept $B_{ij}$ in Table 7 are approximately zero and the simplicity for calculation, we set $B_{ij} = 0$ here, and the $\mu_{kj}$ can be written as

$$\mu_{kj} = T_{ch}\frac{C_{kj}}{\sqrt{s_{NN}}}, \tag{12}$$

where $k = \pi$, $K$, $p$, $u$, $d$, and $s$, $j$ represents different centralities, and $C_{kj}$ is a linear combination of $A_{ij}$, i.e.,

$$
\begin{cases}
C_{\pi j} = -\dfrac{1}{2} A_{\pi j}, \\[2mm]
C_{Kj} = -\dfrac{1}{2} A_{Kj}, \\[2mm]
C_{pj} = -\dfrac{1}{2} A_{pj}, \\[2mm]
C_{uj} = -\dfrac{1}{6} (A_{\pi j} + A_{pj}), \\[2mm]
C_{dj} = -\dfrac{1}{6} (-2A_{\pi j} + A_{pj}), \\[2mm]
C_{sj} = -\dfrac{1}{6} (A_{\pi j} - 2A_{Kj} + A_{pj}).
\end{cases}
\tag{13}
$$

Let $\frac{d\mu_{ij}}{d\sqrt{s_{NN}}} = 0$, and we obtain the energy value ($\sqrt{s_{NN}} = 3.526$ GeV) at the maximum.

It must be emphasized that, due to the lack of data in a low-energy region, the maximum here is only a prediction according to these linear fits, not a certainty. The energy 3.526 GeV at the maximum possibly is the critical energy of phase transition from a liquid-like hadron state to a gas-like quark state in the collision system. At this special energy, the chemical potentials for all cases have the maximum, which indicates that the density of the baryon number has the largest value and the mean-free-path of particles has the smallest value. This means that the hadronic interactions play an important role at this special stage. When the energy is higher than 3.526 GeV, these chemical potentials gradually decrease with the increase of energy, which indicates that the density of baryon number gradually decreases [9], the mean-free-path increases, the shear viscosity over entropy density gradually weakens [44], the hadronic interactions gradually fade, and the partonic interactions gradually become greater. When the energy increases to a very high value, especially the LHC energy, the chemical potentials of all types of particles approach zero, which means that the density of the baryon number and the viscous effect approach zero, and the collision system possibly changes completely from the hadron-dominant state to the quark-dominant state, which denotes the partonic interactions possibly play a dominant role at very high energy [1,45], and the strongly coupled QGP (sQGP) has been observed [5–7].

We must point out that, since the maximum is predicted by the empirical formula, the critical energy value extracted from it has a large fluctuation. In other words, although the fluctuation exists or is even large, it does not mean that the extracted energy value must be wrong. Thus, at the extracted energy, there may be a phase transition critical point. The extracted critical energy (3.526 GeV) of phase transition is consistent with our previous result [21] and the result (below 19.6 GeV) by the STAR Collaboration [1], but less than the result (between 11.5 GeV and 19.6 GeV) of a study based on a correlation between the collision energy and transverse momentum [39,45,46]. Therefore, we still need to make more efforts to find or correct the critical energy point through new methods or theories.

## 4. Summary and Conclusions

The $p_T$ spectra of final-state light flavor particles, $\pi^{\pm}$, $K^{\pm}$, $p$, and $\bar{p}$, produced in Au-Au collisions for different centralities over an energy range from 7.7 to 62.4 GeV are described by a two-component Erlang distribution in the frame of a multi-source thermal model. The fitting results are in agreement with the experimental data recorded by the STAR Collaboration.

The fitting parameters of two-component Erlang $p_T$ distribution shows that, the first component corresponding to a narrow low-$p_T$ region is contributed by the soft excitation process where a few (2–4) sea quarks and gluons take part in, and the second component corresponding to a wide high-$p_T$ region is contributed by the hard scattering process coming from a more violent collision between two valence quarks in incident nucleons. The relative weight factor of a soft excitation process shows that the contribution ratio of

a soft excitation process is more than 60%, which indicates that the excitation degree of collision system is mainly contributed by the soft excitation process.

The yield ratios, $k_\pi$, $k_K$, and $k_p$, of negative to positive particle versus collision energy, and centrality are obtained from the normalization constants. This study shows that, although the dependence of $k_p$ on centrality is higher than that of $k_K$, the dependence of $k_K$ on centrality is higher than that of $k_\pi$, the dependences of the three yield ratios on centrality are not significant, especially for $\pi$. The logarithms of the three types of yield ratios, $\ln(k_\pi)$, $\ln(k_K)$, and $\ln(k_p)$, show obvious linear dependence on $1/\sqrt{s_{NN}}$.

The energy- and centrality-dependent chemical potentials of light hadrons, $\mu_\pi$, $\mu_K$, and $\mu_p$, and quarks, $\mu_u$, $\mu_d$, and $\mu_s$, are extracted from the yield ratios. With the increase in energy over a range from 7.7 to 62.4 GeV, all the chemical potentials (the absolute magnitude for $\pi$) obviously decrease. When the collision energy is very high, all types of chemical potentials are small and tend to be a limiting value of zero. Overall, the dependencies of the six types of energy-dependent chemical potentials on centrality are relatively more obvious in the low energy region than that in the high energy region, but the six energy-dependent chemical potentials in different centrality classes are very close, which indicates that the dependencies of the energy-dependent chemical potentials from Au-Au collisions on centrality are relatively not so significant.

All the derived curves of energy- and centrality-dependent chemical potentials of hadrons and quarks, based on the linear relationships between the logarithms of yield ratios and $1/\sqrt{s_{NN}}$, simultaneously show the maximum (the absolute magnitude for $\pi$) at 3.526 GeV, which is possibly the critical energy of phase transition from a liquid-like hadron state to a gas-like quark state in the collision system, when the density of the baryon number in Au-Au collisions has a large value and the hadronic interactions play an important role. When energy continues to increase, all types of chemical potentials become small, which indicates the density of a baryon number gradually decreases, the mean-free-path gradually increases, and the viscous effect gradually weakens. At this time, the hadronic interactions gradually fade and the partonic interactions gradually become greater. When the energy rises to a very high value, especially to the LHC, all types of chemical potentials tend towards zero, which indicates that the collision system possibly completely changes from the liquid-like hadron-dominant state to the gas-like quark-dominant state when the partonic interactions possibly play a dominant role.

**Author Contributions:** Conceptualization, H.-R.W.; methodology, B.-H.H.; software, H.-Y.W.; validation, X.-W.H. and H.-R.W.; formal analysis, H.-R.W.; investigation, X.-W.H. and H.-R.W.; resources, H.-R.W.; data curation, X.-W.H.; writing—original draft preparation, H.-R.W.; writing—review and editing, W.-T.Z.; visualization, X.-W.H.; supervision, F.-M.W.; project administration, F.-M.W.; funding acquisition, H.-R.W. All authors have read and agreed to the published version of the manuscript.

**Funding:** This work was supported by the National Natural Science Foundation of China under Grant Nos. 11847114.

**Institutional Review Board Statement:** Not applicable.

**Informed Consent Statement:** Not applicable.

**Data Availability Statement:** All data are quoted from the mentioned references. As a phenomenological work, this paper does not report new data.

**Conflicts of Interest:** The authors declare that there are no conflicts of interest regarding the publication of this paper.

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
