# Peer review of "Centrality-Dependent Chemical Potentials of Light Hadrons and Quarks Based on pT Spectrum and Particle Yield Ratio in Au-Au Collisions at RHIC Energies"

_universe, doi:10.3390/universe8080420_

Round 1
Reviewer 1 Report
Referee’s Report on Universe-1819707
„Centrality-dependent chemical potentials …“
by Xing-Wei He et al.
The paper adds a further attempt to extract from measured pT spectra (at midrapidity) of various hadron species their chemical potentials in dependence on centrality and beam energy in heavy-ion collisions at RHIC-BES energies. The key quantity is the employed distribution function, which must be described and motivated with much care (in particular if non-standard parameterizations are used). The authors fail to do so in explaining in a concise manner to the reader the relation (and benefit) of Erlang vs. grand-canonical etc. distributions and their parameters, especially chemical potential and temperature.
Once a successful fit of data is accomplished, one must compare with literature! I see a discrepancy of proton-chemical potential (which is a proxy of the baryo-chemical potential) with well-established results. The local maximum at a certain beam energy let me argue that something in the approach is completely wrong: at low beam energies one expects a continuous rise (corresponding to larger net-baryon densities). This ist to be related to the unprofessionally looking unbelievably precise value of the „critical energy“.
Supposed the authors can overcome my critics, minor remarks are as follows:
- language needs substantial improvement,
- the title should indicate that the paper focuses on RHIC energies,
- Abstract should be compressed (display the message and do not summarize results),
- since some time, species-dependent freeze-out temperatures are discussed (i.e. give reasoning for using one unique),
- Tables 1 – 6 look awfully (maybe, some of their information can be condensed in Figs.).
Reviewer 2 Report
This review covers the article entitled, Centrality-dependent chemical potentials of light hadrons and quarks based on transverse momentum spectrum and particle yield ratio in Au-Au collisions.
Summary
-------
The authors studied the transverse momentum spectra of charged particles (π, K, and p) in Au+Au collisions at different energies from STAR 7.7 – 62.4 GeV [30-31] using a two-component Erlang distribution [14, 23, 24] in the framework of multi-source thermal model. The yield ratios of negative to positive particles are extracted from the normalization constants by fitting the spectra. On the basis of yield ratios, the energy and centrality dependent chemical potentials of light hadrons (π, K, and p) and quarks (u, d, and s) are extracted. The logarithms of these yield ratios show obvious linear dependence on 1/ √ sNN. The extracted chemical potentials (the absolute magnitude for π) of light hadrons and quarks show obvious dependence on energy, and decrease with the increase of energy. The authors also observed that with the increase of energy, all types of chemical potentials become small and tend to zero at very high energy, which indicates that with the increase of energy, the hadronic interactions gradually fade and the partonic interactions become greater, and when the energy rises to a very high value, especially to the LHC, the collision system possibly changes completely from the liquid-like hadron-dominant state to the gas-like quark-dominant state, when the partonic interactions possibly play a dominant role
The authors provide an extensive discussion on data set and analysis method. In summary, the authors point out that the The fitting parameters of two-component Erlang pT distribution shows that, the first component corresponding to a narrow low-pT region is contributed by the soft excitation process where a few (2–4) sea quarks and gluons take part in, and the second component corresponding to a wide high-pT region is contributed by the hard scattering process coming from a more violent collision between two valent quarks in incident nucleons. The relative weight factor of soft excitation process shows that the contribution ratio of soft excitation process is more than 60%, which indicates that the excitation degree of collision system is mainly contributed by the soft excitation process. When the collision energy is very high, all types of chemical potentials are small and tend to be a limiting value of zero. Overall, the dependences of the six types of energy-dependent chemical potentials on centrality are relatively more obvious in low energy region than that in high energy region, but the six energy-dependent chemical potentials in different centrality classes are very close, which indicates that the dependences of the energy-dependent chemical potentials from Au-Au collisions on centrality are relatively not so significant. All the derived curves of energy- and centrality-d
The similar work has already been reported in ref. [18].
Review
------
This paper is well organized and clearly presented. The analysis is straightforward, and there is significant discussion of analysis techniques and contributing errors. The authors provide references of related work.
Major Questions:
The authors already reported the transverse momentum spectra of charged particles (π, K, and p) in Au+Au collisions for only 0-5% centrality at different energies from STAR 7.7 – 62.4 GeV in Ref. [18]. My question is what is the motivation behind studying at other centrality bins? Keeping in mind that authors already reported most central collisions in Adv. High Energy Phys., 2020: 1265090 (2020)
when I compare the table 1 of the current manuscript with the table 1 from the ref. Adv. High Energy Phys., 2020: 1265090 (2020) , some values of 0-5% centrality are different for example , |, which should not be differ by running the code again and again. Please clarify in order to avoid confusion.
Please clarify or cross-check Fig. 8 why the data point of at 20-30% centrality at 62.4 GeV is much higher as compared to others?
Please add the data/model in the lower panel of Fig 1 – 6.
Please improve the Figure 10, for example there is no space between natural log and ( ) on y-axis and make the x-axis label bold
Please make the x-axis bold of figure 11
The authors claimed that the fitting parameters of two-component Erlang pT distribution shows that, the first component corresponding to a narrow low-pT region is contributed by the soft excitation process where a few (2–4) sea quarks and gluons take part in, and the second component corresponding to a wide high-pT region is contributed by the hard scattering process coming from a more violent collision between two valent quarks in incident nucleons. Could you please clarify how you are estimating the few (2-4) sea quarks and gluons?
Minor:
My other concern is related to the writing of the manuscript. The overall text should be improved significantly. Some of the examples are below.
L1: We describe à we analyze
L2: No need to write gold+gold due to the reason that it is well known to the high energy physics community
L2: remove range
Rephrase the sentence starting from “The study shows that …” on L7 for better understanding
Rephrase the sentence starting from “the dependence of energy dependent ….” On L11
Use of gradually is twice on L18
Remove “or quark matter” from L34
Property à properties L34
Critical à possible critical L35
Its à It is L35
Remove process of collision system L43
Transverse momentum ($p_T$) L43
Transverse momentum à $p_T$ L44
Remove “some” and “stage of ” L45
Transverse momentum à $p_T$ L47
There is no space between distribution and reference L48
Transverse momentum à $p_T$ L49, L53,61 and at other places where applicable
The stage of chemical freeze-out à chemical freeze-out stage L50,66, L103 and at other places where applicable
There is no space between distribution and reference L58
Blast-wave modle à Blast-wave model L58
By à From L65
The stage of kinetic freeze-out à kinetic freeze-out stage L68, L100 and at other places where applicable
Resonance à resonances L69
Rephrase the sentence at L78 and remove gold-gold
Obtain à obtained L82
There is no space between distribution and reference L87
Remove “further” L89
Valent or Valence??? L83
Remove “region” L94
Remove comma after “p” L96
Overall, the significant improvement is required in later text before it can be accepted for publication

Reviewer 3 Report
thanks for your nice work, but I have some comments:
1- The authors state work "obvious" around 20 times in the manuscript they should replace it with suitable word.
2- Reference should update the last reference 2020. Please add newer references.
3- Is there any chance to compare your work with LHC data?
LHC had also data with different energy for heavy ions.
4- Some text need to modified as below:
In abstract:
line 10: energy, and => energy and
1. Introduction:
line66: freeze-out according to => freeze-out, according to
line 74: So actually we can => So actually, we can
2. Model and formulism
line 91: a few sea => a few seas
line95: [24-26] and the method => [24-26], and the method
line just above eq. 1: source obey to => source obeys to
line 98: normalization constants corresponding to => normalization constants are corresponding to
line 112: lack of the experimental data =>lack of experimental data
line 114: the hadrons containing =>the hadrons are containing
3- Results and discussion
line137: excitation process are more than => excitation process is more than
line 167: regular trends with increase => regular trends with increasing
line 171: with increase of energy => with increasing of energy
line171: but do not shows obvious => but does not show obvious
The same change in lines 175 and 176 increase => increasing
line199: cross-section of absorbtion => cross-section of absorption
line: 235: maximum possibly => maximum possible
line 252: large, it => large; it
Thanks so much
Round 2
Reviewer 1 Report
The authors decided to keep
(i) the 6 extraordinarily long tables of parameters and
(ii) the quotation of the unbelievably accurate value of \sqrt{s_NN} of a
guessed (but unspecified) "phase transition critical point",
unfortunately and in sharp conflict with my previous recommendations.
Results in Fig.11 seem to contradict, at lower values of \sqrt{s_NN},
the by-now well established systematic (cf. Fig. 9 in [2202.12750],
Fig.2 in [1906.01954], Fig.1 in [1711.02001] to recall a few of the
many, many analyses documented in the contemporary literature).
Otherwise, the paper (amended with a number of improvements in the
revised version) might have some merits w.r.t. the detailed study of
the centrality dependence, even within the employed off-mainstream
framework.
I therefore would not hinder the publication.
Reviewer 2 Report
Thank you very much to the authors for implementing my suggestions. The answers are comprehensive and the physics message is very clear. Therefore, I recommend this paper be published in its current form.